# Single Loop Gaussian Homotopy Method for Non-convex Optimization

**Hidenori Iwakiri**[*]
The University of Tokyo, RIKEN AIP
iwakiri-hidenori2020@g.ecc.u-tokyo.ac.jp

**Yuhang Wang**[*]
The University of Tokyo
utyuuhikou@gmail.com

**Shinji Ito**
NEC Corporation, RIKEN AIP
i-shinji@nec.com

**Akiko Takeda**
The University of Tokyo, RIKEN AIP
takeda@mist.i.u-tokyo.ac.jp

## Abstract

The Gaussian homotopy (GH) method is a popular approach to finding better stationary points for non-convex optimization problems by gradually reducing a parameter value $t$, which changes the problem to be solved from an almost convex one to the original target one. Existing GH-based methods repeatedly call an iterative optimization solver to find a stationary point every time $t$ is updated, which incurs high computational costs. We propose a novel single loop framework for GH methods (SLGH) that updates the parameter $t$ and the optimization decision variables at the same. Computational complexity analysis is performed on the SLGH algorithm under various situations: either a gradient or gradient-free oracle of a GH function can be obtained for both deterministic and stochastic settings. The convergence rate of SLGH with a tuned hyperparameter becomes consistent with the convergence rate of gradient descent, even though the problem to be solved is gradually changed due to $t$. In numerical experiments, our SLGH algorithms show faster convergence than an existing double loop GH method while outperforming gradient descent-based methods in terms of finding a better solution.

## 1 Introduction

Let us consider the following non-convex optimization problem:

$$\underset{x \in \mathbb{R}^d}{\text{minimize}} \quad f(x), \tag{1}$$

where $f : \mathbb{R}^d \to \mathbb{R}$ is a non-convex function. Let us also consider the following stochastic setting:

$$f(x) := \mathbb{E}_\xi[\bar{f}(x; \xi)], \tag{2}$$

where $\xi$ is the random variable following a probability distribution $P$ from which i.i.d. samples can be generated. Such optimization problems attract significant attention in machine learning, and at the same time, the need for optimization algorithms that can find a stationary point with smaller objective value is growing. For example, though it is often said that simple gradient methods can find global minimizers for deep learning (parameter configurations with zero or near-zero training loss), such beneficial behavior is not universal, as noted in [16]; the trainability of neural nets is highly dependent on network architecture design choices, variable initialization, etc. There are also various other highly non-convex optimization problems in machine learning (see e.g., [13]).

---

[*]The first two authors contributed equally.

36th Conference on Neural Information Processing Systems (NeurIPS 2022).

Table 1: Each theorem shows the iteration complexity of SLGH with respect to $\epsilon$ and the dimension of input space $d$ to reach an $\epsilon$-stationary point in the corresponding problem setting. "const. $\gamma$" shows the complexity when we treat the decreasing parameter $\gamma$ as a constant. "tuned $\gamma$" shows the lowest complexity of SLGH attained by updating $t$ appropriately, which matches the complexity of the standard first- or zeroth-order methods (see e.g., Theorem 3.4). We also consider two cases of a zeroth-order setting: "exact $f$", in which we can query the exact or stochastic function value, and "err. $f$", in which we can only access the function value with bounded error.

| | 1) first-order | zeroth-order | |
| | | 2) exact $f$ | 3) err. $f$ |
|---|---|---|---|
| a) deterministic | Thm. 3.4 | Thm. 4.1 | Thm. C.1 |
| const. $\gamma$ | $O\left(\frac{d^{3/2}}{\epsilon^2}\right)$ | $O\left(\frac{d^2}{\epsilon^2}\right)$ | $O\left(\frac{d^3}{\epsilon^2}\right)$ |
| tuned $\gamma$ | $O\left(\frac{1}{\epsilon^2}\right)$ | $O\left(\frac{d}{\epsilon^2}\right)$ | $O\left(\frac{d}{\epsilon^2}\right)$ |
| b) stochastic | Thm. 3.5 | Thm. 4.2 | Thm. C.2 |
| const. $\gamma$ | $O\left(\frac{d}{\epsilon^4} + \frac{d^{3/2}}{\epsilon^2}\right)$ | $O\left(\frac{d^2}{\epsilon^4}\right)$ | $O\left(\frac{d^2}{\epsilon^4} + \frac{d^3}{\epsilon^2}\right)$ |
| tuned $\gamma$ | $O\left(\frac{1}{\epsilon^4}\right)$ | $O\left(\frac{d}{\epsilon^4}\right)$ | $O\left(\frac{d}{\epsilon^4}\right)$ |

The Gaussian homotopy (GH) method is designed to avoid poor stationary points by building a sequence of successively smoother approximations of the original objective function $f$, and it is expected to find a good stationary point with a small objective value for a non-convex problem. More precisely, using the GH function $F(x,t)$ with a parameter $t \geq 0$ that satisfies $F(x,0) = f(x)$, the method starts from solving an almost convex smoothed function $F(x,t_1)$ with some sufficiently large $t_1 \geq 0$ and gradually changes the optimization problem $F(x,t)$ to the original one $f(x)$ while decreasing the parameter $t$. The homotopy method developed so far, then, consists of a double loop structure; the outer loop reduces $t$, and the inner loop solves $\min_x F(x,t)$ for the fixed $t$.

**Related research on the GH method** The GH method is popular owing to its ease of implementation and the quality of its obtained stationary points, i.e., their function values. The nature of this method was first proposed in [2], and it was then successfully applied in various fields, including computer vision [24, 3, 4], physical sciences [12] and computation chemistry [29]. [11] introduces machine learning applications for the GH method, and an application to tuning hyperparameters of kernel ridge regression [25] has recently been introduced. Although there have been recent studies on the GH function $F(x,t)$ [19, 20, 11], all existing GH methods use the double loop approach noted above. Moreover, to the best of our knowledge, there are no existing works that give theoretical guarantee for the convergence rate except for [11]. It characterizes a family of non-convex functions for which a GH algorithm converges to a global optimum and derives the convergence rate to an $\epsilon$-optimal solution. However, the family covers only a small part of non-convex functions, and it is difficult to check whether the required conditions are satisfied for each function. See Appendix A for more discussion on related work.

**Motivation for this work** This paper proposes novel deterministic and stochastic GH methods employing a single loop structure in which the decision variables $x$ and the smoothing parameter $t$ are updated at the same time using individual gradient/derivative information. Using a well-known fact in statistical physics on the relationship between the *heat equation* and Gaussian convolution of $f$, together with *the maximum principle* (e.g., [9]) for the *heat equation*, we can see that a solution $(x^*, t^*)$ minimizing the GH function $F(x,t)$ satisfies $t^* = 0$; thus, $x^*$ is also a solution for (1). This observation leads us to a single loop GH method (SLGH, in short), which updates the current point $(x_k, t_k)$ simultaneously for $\min_{x \in \mathbb{R}^d, t \geq 0} F(x,t)$. The resulting SLGH method can be regarded as an application of the steepest descent method to the optimization problem, with $(x,t)$ as a variable. We are then able to investigate the convergence rate of our SLGH method so as to achieve an $\epsilon$-stationary point of (1) and (2) by following existing theoretical complexity analyses.

We propose two variants of the SLGH method: SLGH$_d$ and SLGH$_r$, which have different update rules for $t$. SLGH$_d$ updates $t$ using the derivative of $F(x,t)$ in terms of $t$, based on the idea of viewing $F(x,t)$ as the objective function with respect to the variable $(x,t)$. Though this approach is effective in finding good solutions (as demonstrated in Appendix D.4), it requires additional computational cost due to the calculation of $\frac{\partial F}{\partial t}$. To avoid this additional computational cost, we also consider

SLGH$_r$ that uses fixed-rate update rule for $t$. We also show that both SLGH$_d$ and SLGH$_r$ have the same theoretical guarantee.

Table 1 summarizes the convergence rate of our SLGH method to reach an $\epsilon$-stationary point under a number of problem settings. Since the convergence rate depends on the decreasing speed of $t$, we list two kinds of complexity in the table; details are described in the caption.

We consider the three settings in which available oracles differ. In Case 1), the full (or stochastic) gradient of $F(x,t)$ in terms of $x$ is available for the deterministic problem (1) (or stochastic problem (2), respectively). However, in this setting, we have to calculate Gaussian convolution for deriving GH functions and their gradient vectors, which becomes expensive, especially for high-dimensional applications, unless closed-form expression of Gaussian convolution is possible. While [18] provides closed-form expression for some specific functions $f$, such as polynomials, Gaussian RBFs, and trigonometric functions, such problem examples are limited. As Case 2), we extend our deterministic and stochastic GH methods to the zeroth-order setting, for which the convolution computation is approximated using only the function values. Another zeroth-order setting, Case 3), is also considered in this paper: the inexact function values (more precisely, the function value with bounded error) can be queried similarly as in the setting in [14]. See Appendix C for more details.

Although no existing studies have analyzed the complexity of a double loop GH method to find an $\epsilon$-stationary point, we can see that its inner loop requires the same complexity as GD (gradient descent) method up to constants. Furthermore, as noted above, the complexity of the SLGH method with a tuned hyperparameter matches that of GD method. Thus, the SLGH method becomes faster than a double loop GH method by around the number of outer loops. The SLGH method is also superior to double loop GH methods from practical perspective, because in order to ensure convergence of their inner loops, we have to set the stepsize conservatively, and furthermore a sufficiently tuned terminate condition must be required.

**Contributions**    We can summarize our contribution as follows:

(1) We propose novel deterministic and stochastic single loop GH (SLGH) algorithms and analyze their convergence rates to an $\epsilon$-stationary point. As far as we know, this is the first analysis of convergence rates of GH methods for general non-convex problems (1) and (2). For non-convex optimization, the convergence rate of SLGH with a tuned hyperparameter becomes consistent with the convergence rate of gradient descent, even though the problem to be solved is gradually changed due to $t$. At this time, the SLGH algorithms become faster than a double loop one by around its number of outer loops.

(2) We propose zeroth-order SLGH (ZOSLGH) algorithms based on zeroth-order estimators of gradient and Hessian values, which are useful when Gaussian smoothing convolution is difficult. We also consider the possibly non-smooth case in which the accessible function contains error, and we derive the upper bound of the error level for convergence guarantee.

(3) We empirically compare our proposed algorithm and other algorithms in experiments, including artificial highly non-convex examples and black-box adversarial attacks. Results show that the proposed algorithm converges much faster than an existing double loop GH method, while it is yet able to find better solutions than are GD-based methods.

## 2    Standard Gaussian homotopy methods

**Notation:**    For an integer $N$, let $[N] := \{1, ..., N\}$. We express $\chi_{[N]} := \{\chi_1, \ldots, \chi_N\}$ for a set of some vectors. We also express the range of the smoothing parameter $t$ as $\mathcal{T} := [0, t_1]$, where $t_1$ is an initial value of the smoothing parameter. Let $\|\cdot\|$ denote the Euclidean norm and $\mathcal{N}(0, \mathrm{I}_d)$ denote the $d$-dimensional standard normal distribution.

Let us first define Gaussian smoothed function.

**Definition 2.1.** Gaussian smoothed function $F(x,t)$ of $f(x)$ is defined as follows:

$$F(x,t) := \mathbb{E}_{u \sim \mathcal{N}(0,\mathrm{I}_d)}[f(x+tu)] = \int f(x+ty)k(y)dy, \tag{3}$$

where $k(y) = (2\pi)^{-d/2} \exp\left(-\|y\|^2/2\right)$ is referred to as the Gaussian kernel.

The idea of Gaussian smoothing is to take an expectation over the function value with a Gaussian distributed random vector $u$. For any $t > 0$, the smoothed function $F(x, t)$ is a $C^\infty$ function, and $t$ plays the role of a smoothing parameter that controls the level of smoothing.

Here, let us show the link between Gaussian smoothing and the *heat equation* [28]. The Gaussian smoothing convolution is basically the solution of the *heat equation* [28].

$$\frac{\partial}{\partial t}\hat{u} = \Delta_x \hat{u}, \quad \hat{u}(\cdot, 0) = f(\cdot), \tag{4}$$

where $\Delta_x$ denotes the Laplacian. The solution of the *heat equation* is $\hat{u}(x, t) = (\frac{1}{4\pi t})^{\frac{d}{2}} \int f(y) e^{-\frac{\|x-y\|^2}{4t}} dy$. This can be made the same as the Gaussian smoothing function $F(x, t)$ by scaling its coefficient, which only changes the speed of progression.

Corollary 9 in [21] shows a sufficient condition for ensuring that $f$ has the asymptotic strict convexity in which the smoothed function $F(x, t)$ becomes convex if a sufficiently large smoothing parameter $t$ is chosen. On this basis, the standard GH method, Algorithm 1, starts with a (almost) convex optimization problem $F(x, t)$ with large parameter value $t \in \mathbb{R}$ and gradually changes the problem toward the target non-convex $f(\cdot) = F(\cdot, 0)$ by decreasing $t$ gradually. [11] reduces $t$ by multiplying by a factor of $1/2$ for each iteration $k$. [20] focuses more on theoretical work w.r.t. the general setting and do not discuss the update rule for $t$.

---

**Algorithm 1** Standard GH method ([20, 11])

---

**Require:** Objective function $f$, iteration number $T$, sequence $\{t_1, \ldots, t_T\}$ satisfying $t_1 > \cdots > t_T$.
    Find a solution $x_1$ for minimizing $F(x, t_1)$.
    **for** $k = 1$ to $T$ **do**
        Find a stationary point $x_{k+1}$ of $F(x, t_{k+1})$ with the initial solution $x_k$.
    **end for**
    **return** $x_T$

---

## 3   Single loop Gaussian homotopy algorithm

A function $h(x)$ is $L_0$-*Lipschitz* with a constant $L_0$ if for any $x, y \in \mathbb{R}^d$, $|h(x) - h(y)| \le L_0 \|x - y\|$ holds. In addition, $h(x)$ is $L_1$-*smooth* with a constant $L_1$ if for any $x, y \in \mathbb{R}^d$, $\|\nabla h(x) - \nabla h(y)\| \le L_1 \|x - y\|$ holds. Let us here list assumptions for developing algorithms with convergence guarantee.
**Assumption A1.**

  (i) Objective function $f$ satisfies $\sup_{x \in \mathbb{R}^d} \mathbb{E}_u[|f(x + tu)|] < \infty$ (In the stochastic setting, $f$ satisfies $\sup_{x \in \mathbb{R}^d, \xi} \mathbb{E}_u[|\bar{f}(x + tu; \xi)|] < \infty$).

  (ii) The optimization problem (1) has an optimal value $f^*$.

  (iii) Objective function $f(x)$ is $L_0$-*Lipschitz* and $L_1$-*smooth* on $\mathbb{R}^d$ (In the stochastic setting, $\bar{f}(x; \xi)$ is $L_0$-*Lipschitz* and $L_1$-*smooth* on $\mathbb{R}^d$ in terms of $x$ for any $\xi$).

Assumption (i) for making $F(x, t)$ well-defined and enabling to exchange the order of differentiation and integration, as well as Assumption (ii), is mandatory for theoretical analysis with the GH method. Assumption (iii) is often imposed for gradient-based methods. This is a regular boundedness and smoothness assumption in recent non-convex optimization analyses (see e.g., [1, 17, 7]).

In the remainder of this section, we consider the nature of the GH method and propose a more efficient algorithm, a SLGH algorithm. We then provide theoretical analyses for our proposed SLGH algorithm.

### 3.1  Motivation

The standard GH algorithm needs to solve an optimization problem for a given smoothing factor $t$ in each iteration and manually reduce $t$, e.g., by multiplying some decreasing factor. To simplify this process, we consider an alternative problem as follows:

$$\underset{x \in \mathbb{R}^d, t \in \mathcal{T}}{\text{minimize}} \quad F(x, t), \tag{5}$$

where $F(x,t)$ is the Gaussian smoothed function of $f(x)$. This single loop structure can reduce the number of iterations by optimizing $x$ and $t$ at the same time.

The following theorem is a (almost) special case of Theorem 6 in [9],[2] which is studied in statistical physics but may not be well-known in machine learning and optimization communities. This theorem shows that the optimal solution of (5) $(x^*, t^*)$ satisfies $t^* = 0$, and thus $x^*$ is also a solution for (1). Therefore, we can regard $F(x,t)$ as an objective function in the SLGH method.

**Theorem 3.1.** *Suppose that Assumptions A1 (i) and (ii) are satisfied. Unless $f$ is constant a.e., the minimum of the GH function $F(x,t)$ will be always found at $t = 0$, and the corresponding $x$ will be an optimal solution for* (1).

We present a proof of this theorem in Appendix B.1. The proof becomes much easier than that in [9] due to its considering a specific case.

Let us next introduce an update rule for $t$ utilizing the derivative information. When we solve the problem (5) using a gradient descent method, the update rule for $t$ becomes $t_{k+1} = t_k - \eta \frac{\partial F}{\partial t}$, where $\eta$ is a step size. The formula (4) in the *heat equation* implies that the derivative $\frac{\partial F}{\partial t}$ is equal to the Laplacian $\Delta_x F$, i.e., $\frac{\partial F}{\partial t} = \text{tr}(\text{H}_F(x))$, where $\text{H}_F(x)$ is the Hessian of $F$ in terms of $x$. Since $\text{tr}(\text{H}_F(x))$ represents the sharpness of minima [8], this update rule can sometimes decrease $t$ quickly around a minimum and find a better solution. See Appendix D.4 for an example of such a problem.

## 3.2 SLGH algorithm

Let us next introduce our proposed SLGH algorithm, which has two variants with different update rules for $t$: SLGH with a fixed-ratio update rule (SLGH$_r$) and SLGH with a derivative update rule (SLGH$_d$). SLGH$_r$ updates $t$ by multiplying a decreasing factor $\gamma$ (e.g., 0.999) at each iteration. In contrast to this, SLGH$_d$ updates $t$ while using derivative information. Details are described in Algorithm 2. Algorithm 2 transforms a double loop Algorithm 1 into a single loop algorithm. This single loop structure can significantly reduce the number of iterations while ensuring the advantages of the GH method.

---

**Algorithm 2** Deterministic/Stochastic Single Loop GH algorithm (SLGH)

**Require:** Iteration number $T$, initial solution $x_1$, initial smoothing parameter $t_1$, step size $\beta$ for $x$, step size $\eta$ for $t$, decreasing factor $\gamma \in (0,1)$, sufficient small positive value $\epsilon$

   **for** $k = 1$ to $T$ **do**

$$x_{k+1} = x_k - \beta \widehat{G}_x, \ \widehat{G}_x = \begin{cases} \nabla_x F(x_k, t_k) \ (\text{determ.}) \\ \nabla_x \bar{F}(x_k, t_k; \xi_k), \ \xi_k \sim P \ (\text{stoc.}) \end{cases}$$

$$t_{k+1} = \begin{cases} \gamma t_k \ (\text{SLGH}_r) \\ \max\{\min\{t_k - \eta \widehat{G}_t, \ \gamma t_k\}, \ \epsilon'\} \ (\text{SLGH}_d) \end{cases}, \ \widehat{G}_t = \begin{cases} \frac{\partial F(x_k, t_k)}{\partial t} \ (\text{determ.}) \\ \frac{\partial \bar{F}(x_k, t_k; \xi_k)}{\partial t}, \ \xi_k \sim P \ (\text{stoc.}) \end{cases}$$

   **end for**

---

In the stochastic setting of (2), the gradient of $F(x,t)$ in terms of $x$ is approximated by $\nabla_x \bar{F}(x,t;\xi)$ with randomly chosen $\xi$, where $\bar{F}(x,t;\xi)$ is the GH function of $\bar{f}(x;\xi)$. Likewise, the derivative of $F(x,t)$ in terms of $t$ is approximated by $\frac{\partial \bar{F}(x,t;\xi)}{\partial t}$. The stochastic algorithm in Algorithm 2 uses one sample $\xi_k$. We can extend the stochastic approach to a minibatch one by approximating $\nabla_x F(x,t)$ by $\frac{1}{M} \sum_{i=1}^{M} \nabla_x \bar{F}(x,t;\xi_i)$ with samples $\{\xi_1, \ldots, \xi_M\}$ of some batch size $M$, but for the sake of simplicity, we here assume one sample in each iteration. In this setting, the gradient complexity matches the iteration complexity; thus, we also use the term "iteration complexity" in the stochastic setting. Other methods, such as momentum-accelerated method [27] and Adam [15] can also be applied here. According to Theorem 3.1, the final smoothing parameter needs to be zero. Thus, we

---

[2]Although the assumptions in Theorem 3.1 are stronger than those in the theorem proved by Evans, the statement of ours is also stronger than that of his theorem, in a sense that our theorem guarantees that all optimal solutions satisfy $t = 0$.

multiply $\gamma$ by $t$ even in SLGH$_d$ when the decrease of $t$ is insufficient. We also assure that $t$ is larger than a sufficiently small positive value $\epsilon' > 0$ during an update to prevent $t$ from becoming negative.

### 3.3 Convergence analysis for SLGH

Let us next analyze the worst-case iteration complexity for both deterministic and stochastic SLGHs, but, before that, let us first show some properties for Gaussian smoothed function $F(x, t)$ under Assumption A1 for the original function $f(x)$. In the complexity analyses in this paper, we always assume that $\gamma$ is bounded from above by a universal constant $\bar{\gamma} < 1$, which implies $1/(1-\gamma) = O(1)$.

**Lemma 3.2.** *Let $f(x)$ be a $L_0$-Lipschitz function. Then, for any $t > 0$, its Gaussian smoothed function $F(x, t)$ will then also be $L_0$-Lipschitz in terms of $x$. Let $f(x)$ be a $L_1$-smooth function. Then, for any $t > 0$, $F(x, t)$ will also be $L_1$-smooth in terms of $x$.*

Lemma 3.2 indicates that Assumption A1 given to the function $f(x)$ also guarantees the same properties for $F(x, t)$. Below, we give some bounds between the smoothed function $F(x, t)$ and the original function $f(x)$.

**Lemma 3.3.** *Let $f$ be a $L_0$-Lipschitz function. Then, for any $x \in \mathbb{R}^d$, $F(x, t)$ is also $L_0\sqrt{d}$-Lipschitz in terms of $t$, i.e., for any $x$, smoothing parameter values $t_1, t_2 > 0$, we have $|F(x, t_1) - F(x, t_2)| \leq L_0\sqrt{d}|t_1 - t_2|$.*

On the basis of Lemmas 3.2 and 3.3, the convergence results of our deterministic and stochastic SLGH algorithms can be given as in Theorems 3.4 and 3.5, respectively. Proofs of the following theorems are given in Appendix B.2. Let us first deal with the deterministic setting.

**Theorem 3.4** (**Convergence of SLGH, Deterministic setting**). *Suppose Assumption A1 holds , and let $\hat{x} := x_{k'}$, $k' = \operatorname{argmin}_{k \in [T]} \|\nabla f(x_k)\|$. Set the stepsize for $x$ as $\beta = 1/L_1$. Then, for any setting of the parameter $\gamma$, $\hat{x}$ satisfies $\|\nabla f(\hat{x})\| \leq \epsilon$ with the iteration complexity of $T = O\left(d^{3/2}/\epsilon^2\right)$. Further, if we choose $\gamma \leq d^{-\Omega(\epsilon^2)}$, the iteration complexity can be bounded as $T = O(1/\epsilon^2)$.*

This theorem indicates that if we choose $\gamma$ close to 1, then the iteration complexity can be $O\left(d^{3/2}/\epsilon^2\right)$, which is $O(d^{3/2})$ times larger than the $O(1/\epsilon^2)$-iteration complexity by the standard gradient descent methods [22]. However, we can remove this dependency on $d$ to obtain an iteration complexity matching that of the standard gradient descent, by choosing $\gamma \leq d^{-\Omega(\epsilon^2)}$, as shown in Theorem 3.4. Empirically, settings of $\gamma$ close to 1, e.g., $\gamma = 0.999$, seem to work well enough, as demonstrated in Section 5.

An inner loop of the double loop GH method using the standard GD requires the same complexity as the standard GD method up to constants since the objective smoothed function of inner optimization problem is $L_1$-smooth function. By considering the above results, we can see that the SLGH algorithm becomes faster than the double loop one by around the number of outer loops.

To provide theoretical analyses in the stochastic setting, we need additional standard assumptions.

**Assumption A2.**

(i) The stochastic function $\bar{f}(x; \xi)$ becomes an unbiased estimator of $f(x)$. That is, for any $x \in \mathbb{R}^d$, $f(x) = \mathbb{E}_\xi[\bar{f}(x; \xi)]$ holds.

(ii) For any $x \in \mathbb{R}^d$, the variance of the stochastic gradient oracle is bounded as $\mathbb{E}_\xi[\|\nabla_x \bar{f}(x; \xi) - \nabla f(x)\|^2] \leq \sigma^2$. Here, the expectation is taken w.r.t. random vectors $\{\xi_k\}$.

The following theorem shows the convergence rate in the stochastic setting.

**Theorem 3.5** (**Convergence of SLGH, Stochastic setting**). *Suppose Assumptions A1 and A2 hold. Take $k_1 := \Theta(1/\epsilon^4)$ and $k_2 := O\left(\log_\gamma \min\{d^{-1/2}, d^{-3/2}\epsilon^{-2}\}\right)$ and define $k_0 = \min\{k_1, k_2\}$. Let $\hat{x} := x_{k'}$, where $k'$ is chosen from a uniform distribution over $\{k_0 + 1, k_0 + 2, \ldots, T\}$. Set the stepsize for $x$ as $\beta = \min\left\{1/L_1, 1/\sqrt{T-k_0}\right\}$. Then, for any setting of the parameter $\gamma$, $\hat{x}$ satisfies $\mathbb{E}[\|\nabla f(\hat{x})\|] \leq \epsilon$ with the iteration complexity of $T = O\left(d/\epsilon^4 + d^{3/2}/\epsilon^2\right)$ where the expectation is taken w.r.t. random vectors $\{\xi_k\}$. Further, if we choose $\gamma \leq (\max\{d^{1/2}, d^{3/2}\epsilon^2\})^{-\Omega(\epsilon^4)}$, the iteration complexity can be bounded as $T = O(1/\epsilon^4)$.*

We note that the iteration complexity of $T = O(1/\epsilon^4)$ for sufficiently small $\gamma$ matches that for the standard stochastic gradient descent (SGD) shown, e.g., by [10].

# 4 Zeroth-order single loop Gaussian homotopy algorithm

In this section, we introduce a zeroth-order version of the SLGH algorithms. This ZOSLGH algorithm is proposed for those optimization problems in which Gaussian smoothing convolution is difficult to compute, or in which only function values can be queried.

## 4.1 ZOSLGH algorithm

For cases in which only function values are accessible, approximations for the gradient in terms of $x$ and derivative in terms of $t$ are needed. [23] has shown that the gradient of the smoothed function $F(x, t)$ can be represented as

$$\nabla_x F(x, t) = \frac{1}{t} \mathbb{E}_u([f(x + tu) - f(x)]u), \ u \sim \mathcal{N}(0, \mathrm{I}_d). \tag{6}$$

Thus, the gradient $\nabla_x F(x, t)$ can be approximated by an unbiased estimator $\tilde{g}_x(x, t; u)$ as

$$\tilde{g}_x(x, t; u) := \frac{1}{t}(f(x + tu) - f(x))u, \ u \sim \mathcal{N}(0, \mathrm{I}_d). \tag{7}$$

The derivative $\frac{\partial F}{\partial t}$ is equal to the trace of the Hessian of $F(x, t)$ because the Gaussian smoothed function is the solution of the *heat equation* $\frac{\partial F}{\partial t} = \mathrm{tr}(\mathrm{H}_F(x))$. We can estimate $\mathrm{tr}(\mathrm{H}_F(x))$ on the basis of the second order Stein's identity [26] as follows:

$$\mathrm{H}_F(x) \approx \frac{(vv^\top - \mathrm{I}_d)}{t^2}(f(x + tv) - f(x)), \ v \sim \mathcal{N}(0, \mathrm{I}_d). \tag{8}$$

Thus, the estimator for derivative can be written as:

$$\tilde{g}_t(x, t; v) := \frac{(v^\top v - d)(f(x + tv) - f(x))}{t^2}, \ v \sim \mathcal{N}(0, \mathrm{I}_d). \tag{9}$$

As for the stochastic setting, $f(x)$ in (7) and (9) is replaced by the stochastic function $\bar{f}(x; \xi)$ with some randomly chosen sample $\xi$. The gradient $\nabla_x \bar{F}(x, t; \xi)$ of its GH function $\bar{F}(x, t; \xi)$ can then be approximated by $\tilde{G}_x(x, t; \xi, u) := \frac{\bar{f}(x+tu;\xi) - \bar{f}(x;\xi)}{t}u$, and the derivative $\frac{\partial \bar{F}}{\partial t}$ can be approximated by $\tilde{G}_t(x, t; \xi, v) := \frac{(v^\top v - d)(\bar{f}(x+tv;\xi) - \bar{f}(x;\xi))}{t^2}$ (see Algorithm 3 for more details).

---

**Algorithm 3** Deterministic/Stochastic Zeroth-Order Single Loop GH algorithm (ZOSLGH)

---

**Require:** Iteration number $T$, initial solution $x_1$, initial smoothing parameter $t_1$, step size $\beta$ for $x$, step size $\eta$ for $t$, decreasing factor $\gamma \in (0, 1)$, sufficient small positive value $\epsilon$
  **for** $k = 1$ to $T$ **do**
    Sample $u_k$ from $\mathcal{N}(0, \mathrm{I}_d)$
    $x_{k+1} = x_k - \beta \bar{G}_{x,u}, \ \bar{G}_{x,u} = \begin{cases} \tilde{g}_x(x_k, t_k; u_k) \ \text{(determ.)} \\ \tilde{G}_x(x_k, t_k; \xi_k, u_k), \ \xi_k \sim P \ \text{(stoc.)} \end{cases}$
    Sample $v_k$ from $\mathcal{N}(0, \mathrm{I}_d)$
    $t_{k+1} = \begin{cases} \gamma t_k \ (\text{SLGH}_\mathrm{r}) \\ \max\{\min\{t_k - \eta \bar{G}_{t,v}, \gamma t_k\}, \epsilon'\} \ (\text{SLGH}_\mathrm{d}) \end{cases}, \ \bar{G}_{t,v} = \begin{cases} \tilde{g}_t(x_k, t_k; v_k) \ \text{(determ.)} \\ \tilde{G}_t(x_k, t_k; \xi_k, v_k), \ \xi_k \sim P \ \text{(stoc.)} \end{cases}$
  **end for**

---

## 4.2 Convergence analysis for ZOSLGH

We can analyze the convergence results using concepts similar to those used with the first-order SLGH algorithm. Below are the convergence results for ZOSLGH in both the deterministic and stochastic

settings. Proofs of the following theorems are given in Appendix B.3, and the definitions of $\hat{x}$ are provided in the proofs. We start from the deterministic setting, which is aimed at the deterministic problem (1).

**Theorem 4.1** (**Convergence of ZOSLGH, Deterministic setting**). *Suppose Assumption A1 holds. Take $k_1 := \Theta(d/\epsilon^2)$ and $k_2 := O\left(\log_\gamma d^{-1/2}\right)$, and define $k_0 = \min\{k_1, k_2\}$. Let $\hat{x} := x_{k'}$, where $k'$ is chosen from a uniform distribution over $\{k_0 + 1, k_0 + 2, \ldots, T\}$. Set the stepsize for $x$ as $\beta = 1/(2(d+4)L_1)$. Then, for any setting of the parameter $\gamma$, $\hat{x}$ satisfies $\mathbb{E}[\|\nabla f(\hat{x})\|] \leq \epsilon$ with the iteration complexity of $T = O(d^2/\epsilon^2)$, where the expectation is taken w.r.t. random vectors $\{u_k\}$ and $\{v_k\}$. Further, if we choose $\gamma \leq d^{-\Omega(\epsilon^2/d)}$, the iteration complexity can be bounded as $T = O(d/\epsilon^2)$.*

This complexity of $O(d/\epsilon^2)$ for $\gamma \leq d^{-\Omega(\epsilon^2/d)}$ matches that of zeroth-order GD (ZOGD) [23].

Let us next introduce the convergence result for the stochastic setting. As shown in [10], if we take the expectation for our stochastic zeroth-order gradient oracle with respect to both $\xi$ and $u$, under Assumption A2 (i), we will have

$$\mathbb{E}_{\xi,u}[\tilde{G}_x(x, t; \xi, u)] = \mathbb{E}_u[\mathbb{E}_\xi[\tilde{G}_x(x, t; \xi, u)|u]] = \nabla_x F(x, t).$$

Therefore, $\zeta_k := (\xi_k, u_k)$ behaves similarly to $u_k$ in the deterministic setting.

**Theorem 4.2** (**Convergence of ZOSLGH, Stochastic setting**). *Suppose Assumptions A1 and A2 hold. Take $k_1 := \Theta(d/\epsilon^4)$ and $k_2 := O\left(\log_\gamma d^{-1/2}\right)$, and define $k_0 = \min\{k_1, k_2\}$. Let $\hat{x} := x_{k'}$, where $k'$ is chosen from a uniform distribution over $\{k_0 + 1, k_0 + 2, \ldots, T\}$. Set the stepsize for $x$ as $\beta = \min\{\frac{1}{2(d+4)L_1}, \frac{1}{\sqrt{(T-k_0)(d+4)}}\}$. Then, for any setting of the parameter $\gamma$, $\hat{x}$ satisfies $\mathbb{E}[\|\nabla f(\hat{x})\|] \leq \epsilon$ with the iteration complexity of $T = O(d^2/\epsilon^4)$, where the expectation is taken w.r.t. random vectors $\{u_k\}$, $\{v_k\}$, and $\{\xi_k\}$. Further, if we choose $\gamma \leq d^{-\Omega(\epsilon^4/d)}$, the iteration complexity can be bounded as $T = O(d/\epsilon^4)$.*

This complexity of $O(d/\epsilon^4)$ for $\gamma \leq d^{-\Omega(\epsilon^4/d)}$ also matches that of ZOSGD [10].

## 5 Experiments

In this section, we present our experimental results. We conducted two experiments. The first was to compare the performance of several algorithms including the proposed ones, using test functions for optimization. We were able to confirm the effectiveness and versatility of our SLGH methods for highly non-convex functions. We also created a toy problem in which ZOSLGH$_d$, which utilizes the derivative information $\frac{\partial F}{\partial t}$ for the update of $t$, can decrease $t$ quickly around a minimum and find a better solution than that with ZOSLGH$_r$. The second experiment was to generate examples for a black-box adversarial attack with different zeroth-order algorithms. The target models were well-trained DNNS for CIFAR-10 and MNIST, respectively. All experiments were conducted using Python and Tensorflow on Intel Xeon CPU and NVIDIA Tesla P100 GPU. We show the results of only the adversarial attacks due to the space limitations; other results are given in Appendix D.

**Generation of per-image black-box adversarial attack example.** Let us consider the unconstrained black-box attack optimization problem in [6], which is given by

$$\underset{x \in \mathbb{R}^d}{\text{minimize}} \ f(x) := \lambda \ell(0.5\tanh(\tanh^{-1}(2a) + x)) + \|0.5\tanh(\tanh^{-1}(2a) + x) - a\|^2,$$

where $\lambda$ is a regularization parameter, $a$ is the input image data, and $tanh$ is the element-wise operator which helps eliminate the constraint representing the range of adversarial examples. The first term $\ell(\cdot)$ of $f(x)$ is the loss function for the untargeted attack in [5], and the second term $L_2$ distortion is the adversarial perturbation (the lower the better). The goal of this problem is to find the perturbation that makes the loss $\ell(\cdot)$ reach its minimum while keeping $L_2$ distortion as small as possible. The initial adversarial perturbation $x_0$ was set to 0. We say a successful attack example has been generated when the loss $\ell(\cdot)$ is lower than the attack confidence (e.g., $1e - 10$).

Let us here compare our algorithms, ZOSLGH$_r$ and ZOSLGH$_d$, to three zeroth-order algorithms: ZOSGD [10], ZOAdaMM [6], and ZOGradOpt [11]. ZOGradOpt is a homotopy method with a

double loop structure. In contrast to this, ZOSGD and ZOAdaMM are SGD-based zeroth-order methods and thus do not change the smoothing parameter during iterations.

Table 2 and Figure 1 show results for our experiment. We can see that SGD-based algorithms are able to succeed in the first attack with far fewer iterations than our GH algorithms (e.g., Figure 1(a), Figure 1(d)). Accordingly, the value of $L_2$ distortion decreases slightly more than GH methods. However, SGD-based algorithms have lower success rates than do our SLGH algorithms. This is because SGD-based algorithms remain around a local minimum $x = 0$ when it is difficult to attack, while GH methods can escape the local minima due to sufficient smoothing (e.g., Figure 1(b), Figure 1(e)). Thus, the SLGH algorithms are, on average, able to decrease total loss over that with SGD-based algorithms. In a comparison within GH methods, ZOGradOpt requires more than 6500 iterations to succeed in the first attack due to its double loop structure (e.g., Figure 1(c), Figure 1(f)). In contrast to this, our SLGH algorithms achieve a high success rate with far fewer iterations. Please note that SLGH$_d$ takes approximately twice the computational time per iteration than the other algorithms because it needs additional queries for the computation of the derivative in terms of $t$. See Appendix E for a more detailed presentation of the experimental setup and results.

Table 2: Performance of a per-image attack over 100 images of CIFAR-10 under $T = 10000$ iterations. "Succ. rate" indicates the ratio of success attack, "Avg. iters to 1st succ." is the average number of iterations to reach the first successful attack , "Avg. $L_2$ (succ.)" is the average of $L_2$ distortion taken among successful attacks, and "Avg. total loss" is the average of total loss $f(x)$ over 100 samples. Please note that the standard deviations are large since the attack difficulty varies considerably from sample to sample.

| | Methods | Succ. rate | Avg. iters to 1st succ. | Avg. $L_2$ (succ.) | Avg. total loss |
|---|---|---|---|---|---|
| SGD algo. | ZOSGD | 88% | $\mathbf{835} \pm 1238$ | $0.076 \pm 0.085$ | $27.70 \pm 74.80$ |
| | ZOAdaMM | 85% | $3335 \pm 2634$ | $\mathbf{0.050} \pm 0.055$ | $20.24 \pm 62.48$ |
| GH algo. | ZOGradOpt | 65% | $6789 \pm 1901$ | $0.249 \pm 0.159$ | $41.45 \pm 76.04$ |
| | ZOSLGH$_r$ ($\gamma = 0.999$) | **93%** | $4979 \pm 756$ | $0.246 \pm 0.178$ | $\mathbf{14.26} \pm 54.61$ |
| | ZOSLGH$_d$ ($\gamma = 0.999$) | **92%** | $4436 \pm 805$ | $0.150 \pm 0.084$ | $\mathbf{16.49} \pm 58.69$ |

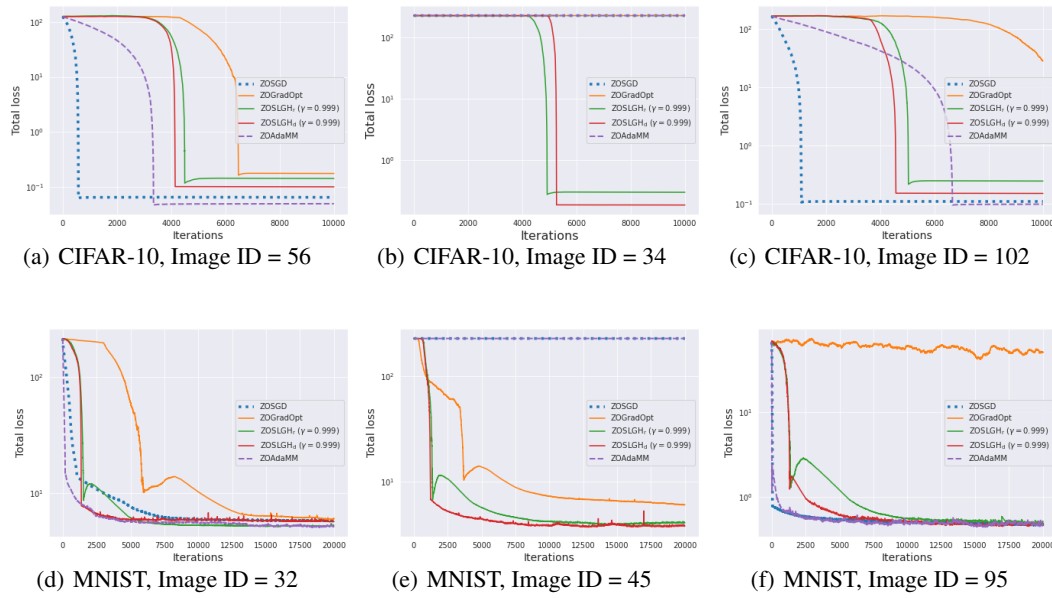

(a) CIFAR-10, Image ID = 56    (b) CIFAR-10, Image ID = 34    (c) CIFAR-10, Image ID = 102

(d) MNIST, Image ID = 32    (e) MNIST, Image ID = 45    (f) MNIST, Image ID = 95

Figure 1: Total loss for generating per-image black-box adversarial examples for different images of CIFAR-10 and MNIST (log scale).

## 6  Summary and future work

We have presented here the deterministic/stochastic SLGH and ZOSLGH algorithms as well as their convergence results. They have been designed for the purpose of finding better solutions with fewer iterations by simplifying the homotopy process into a single loop. We consider this work to be a first attempt to improve the standard GH method.

Although this study has considered the case in which the accessible function contains some error and is possibly non-smooth, we assume the underlying objective function to be smooth. Further work should be carried out to investigate the case in which the objective function itself is non-smooth.

**Acknowledgements**   This work was supported by JSPS KAKENHI Grant Number 19H04069, JST ACT-I Grant Number JPMJPR18U5, and JST ERATO Grant Number JPMJER1903.

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
