for the $\sigma$-nice function. The framework of $\sigma$-nice imposes the two conditions: (i) the solution obtained in each inner loop is located sufficiently close to an optimal solution of the optimization problem in the next inner loop; (ii) the optimization problem in each inner loop is strongly convex around its optimal solutions. Unfortunately, it is not obvious whether we can efficiently judge a function is "$\sigma$-nice", and we cannot apply the analysis results to general non-convex functions. On the other hand, this work tackles a problem of different nature from [14] since it analyzes the convergence rate to an $\epsilon$-stationary point for general non-convex functions.

**Guarantee for the value of the objective function**    [24] provided an upper bound on the objective value attained by a homotopy method. The bound was characterized by a quantity that they referred to as "optimization complexity", which can be analytically computed when the objective function is expressed in some suitable basis functions such as Gaussian RBFs.

**Other smoothing methods**    Smoothing methods other than Gaussian smoothing include [7, 8]. The smoothing kernel in those works is simpler but restricted to specific problem settings. For example, [8] constructs smoothing approximations for optimization problems that can be reformulated by using the plus function $(t)_+ := \max\{0, t\}$.

**Zeroth-order techniques**    In problem settings in which the explicit gradient of the objective function cannot be calculated but the exact function values can be queried, zeroth-order optimization has become increasingly popular due to its potential for wide application. Such a class of applications appears in black-box adversarial attacks on deep neural networks [9], structured prediction [31], and reinforcement learning [36]. Various zeroth-order methods (ZOSGD [13], ZOAdaMM [9], ZOSVRG [20]) have been proposed for such black-box situations. All of them have been developed from ZOGD in [28], which introduces random gradient-free oracles based on Gaussian smoothing with fixed $t$. This trend also applies to research on the GH method. [14] developed a GH method in the zeroth-order setting for which the objective is only accessible through a noisy value oracle. [30] proposed a GH method for hyperparameter tuning based on [14] using two-point zeroth-order estimators [28].

# B  Proofs for theorems and lemmas in Sections 3 and 4

**Notation:** We sometimes denote the expectation with respect to random variables $\chi_{S+1}, \ldots, \chi_T$ $(S, T \in \mathbb{N}, T > S)$ as $\mathbb{E}_\chi[\cdot]$ for the sake of simplicity.

## B.1  Theorem 3.1

**Proof for Theorem 3.1:** Since the optimization problem (1) has an optimal value $f^*$ by Assumption A1 (ii), for any $t \in \mathcal{T}$ and for any $x \in \mathbb{R}^d$, we have

$$F(x, t) - f^* = \mathbb{E}_u[f(x + tu) - f^*] \geq 0.$$

Together with the relationship $F(x, 0) = f(x)$, for any $x \in \mathbb{R}^d$, for any $t \in \mathcal{T}$ and for any optimal solution $x^* \in \mathbb{R}^d$ of the optimization problem (1), we have $F(x, t) - F(x^*, 0) \geq 0$. Furthermore, if we exclude cases where $f(x)$ is constant (a.e.), for any $(x, t) \in \mathbb{R}^d \times \mathcal{T} \setminus \{(x, 0) \mid f(x) = f(x^*)\}$, we obtain

$$F(x, t) - f^* = \mathbb{E}_u[f(x + tu) - f^*] > 0.$$

---

[3]Their method is not exactly a GH method because it smooths the objective function using random variables sampled from the unit ball (or the unit sphere in a zeroth-order setting) rather than Gaussian random variables. However, for the sake of simplicity, we treat it as a GH method in this paper.

Therefore, a minimum of the optimization problem of the GH function $\min_{x \in \mathbb{R}^d, t \in \mathcal{T}} F(x, t)$ holds only at $t = 0$ and the corresponding $x$ becomes an optimal solution of the original optimization problem $\min_{x \in \mathbb{R}^d} f(x)$. $\qquad\square$

## B.2 First-order SLGH algorithm

At the beginning of the subsection, we introduce a lemma that gives upper bounds for moments of Gaussian random variables, and then prove the two lemmas which appeared in the main paper.

**Lemma B.1 (Lemma 1 in [28]).** *Let $u \in \mathbb{R}^d$ be a standard normal random variable. For $p \in [0, 2]$, we have $\mathbb{E}_u[\|u\|^p] \leq d^{p/2}$. If $p \geq 2$, $\mathbb{E}_u[\|u\|^p] \leq (d + p)^{p/2}$ holds.*

**Proof for Lemma 3.2:** According to the definition of Gaussian smoothing in the main paper, we have

$$|F(x, t) - F(y, t)| = \left| \int (f(x + tz)k(z) - f(y + tz)k(z))dz \right|$$

$$\leq \int |f(x + tz) - f(y + tz)| \, k(z)dz$$

$$\leq \int L_0 \|x - y\| k(z)dz$$

$$\leq L_0 \|x - y\|.$$

The proof of $L_1$-*smooth* is similar to that of $L_0$-*Lipschitz*:

$$|\nabla_x F(x, t) - \nabla_x F(y, t)| \leq \int |\nabla f(x + tz) - \nabla f(y + tz)| k(z)dz$$

$$\leq \int L_1 \|x - y\| k(z)dz$$

$$\leq L_1 \|x - y\|.$$

$\qquad\square$

The lemma has proved that the Lipschitz constants of $F(x, t)$ and $\nabla_x F(x, t)$ in terms of $x$ are smaller than those of $f(x)$ and $\nabla f(x)$, respectively. Therefore we can use the Lipschitz constants $L_0$ and $L_1$ of $f(x)$ and $\nabla f(x)$ for $F(x, t)$ and $\nabla_x F(x, t)$.

**Proof for Lemma 3.3:**

$$|F(x, t_1) - F(x, t_2)| = |\mathbb{E}_u[f(x + t_1 u) - f(x + t_2 u)]|$$

$$\leq \mathbb{E}_u[|f(x + t_1 u) - f(x + t_2 u)|]$$

$$\leq \mathbb{E}_u[L_0 |t_1 - t_2| \|u\|]$$

$$\leq L_0 |t_1 - t_2| \sqrt{d},$$

where the last inequality holds due to Lemma B.1. $\qquad\square$

Before going to the convergence theorems, we introduce an additional useful lemma to estimate the gap between the gradient of the smoothed function and the true gradient.

**Lemma B.2.** *Let $f$ be a $L_1$-smooth function.*
*(i) (**Lemma 4 in [28]**) For any $x \in \mathbb{R}^d$ and $t > 0$, we have*

$$\|\nabla f(x)\|^2 \leq 2\|\nabla_x F(x, t)\|^2 + \frac{t^2}{2} L_1^2 (d + 6)^3.$$

*(ii) Further, if $f$ is $L_0$-Lipschitz, for any $x \in \mathbb{R}^d$ and $t > 0$, we have*

$$\|\nabla f(x)\|^2 \leq \|\nabla_x F(x, t)\|^2 + t L_0 L_1 (d + 3)^{3/2}.$$

**Proof for (ii):** We have

$$\|\nabla f(x)\|^2 - \|\nabla_x F(x, t)\|^2 = (\|\nabla f(x)\| + \|\nabla_x F(x, t)\|)(\|\nabla f(x)\| - \|\nabla_x F(x, t)\|)$$

$$\leq 2L_0(\|\nabla f(x)\| - \|\nabla_x F(x, t)\|)$$

$$\leq 2L_0 \|\nabla_x F(x, t) - \nabla f(x)\|.$$

The term $\|\nabla_x F(x, t) - \nabla f(x)\|$ can be upper bounded as follows:

$$\|\nabla_x F(x, t) - \nabla f(x)\| \leq \left\| \mathbb{E}_u \left[ \left( \frac{f(x + tu) - f(x)}{t} - \langle \nabla f(x), u \rangle \right) u \right] \right\|$$

$$\leq \mathbb{E}_u \left[ \left| \frac{1}{t} \left( f(x + tu) - f(x) - t\langle \nabla f(x), u \rangle \right) \right| \|u\| \right]$$

$$\leq \mathbb{E}_u \left[ \frac{tL_1}{2} \|u\|^3 \right]$$

$$\leq \frac{tL_1}{2} (d + 3)^{3/2},$$

where the last second inequality follows from a property of $L_1$-smooth function ($\forall x, y \in \mathbb{R}^d$, $|f(y) - f(x) - \langle \nabla f(x), y - x \rangle| \leq \frac{L_1}{2} \|y - x\|^2$), and the last inequality holds due to Lemma B.1. Therefore, we obtain

$$\|\nabla f(x)\|^2 \leq \|\nabla_x F(x, t)\|^2 + tL_0 L_1 (d + 3)^{3/2}.$$

Now, we are ready to prove Theorem 3.4.

**Proof for Theorem 3.4:** We follow the convergence analysis of gradient descent. According to Assumption A1 and Lemma 3.2, $F(x, t)$ is $L_0$-$Lipschitz$ and $L_1$-$smooth$ in terms of $x$. Therefore, we have

$$F(x_{k+1}, t_k) \leq F(x_k, t_k) + \langle \nabla_x F(x_k, t_k), (x_{k+1} - x_k) \rangle + \frac{L_1}{2} \|x_{k+1} - x_k\|^2$$

$$= F(x_k, t_k) - \left( \beta - \frac{L_1}{2} \beta^2 \right) \|\nabla_x F(x_k, t_k)\|^2,$$

where the last equation holds due to the updating rule of the gradient descent: $x_{k+1} - x_k = -\beta \nabla_x F(x_k, t_k)$. Then, we can get the upper bound for $\|\nabla_x F(x, t)\|^2$:

$$\left( \beta - \frac{L_1}{2} \beta^2 \right) \|\nabla_x F(x_k, t_k)\|^2 \leq F(x_k, t_k) - F(x_{k+1}, t_k)$$

$$= F(x_k, t_k) - F(x_{k+1}, t_{k+1}) + F(x_{k+1}, t_{k+1}) - F(x_{k+1}, t_k)$$

$$\leq F(x_k, t_k) - F(x_{k+1}, t_{k+1}) + L_0 |t_{k+1} - t_k| \sqrt{d},$$

where the last inequality follows from Lemma 3.3.

Now, sum up the above inequality for all iterations $k_0 + 1 \leq k \leq T$ ($T > k_0 \in \mathbb{N}$), and denote the minimum of $f$ as $f^*$, then we have

$$\left( \beta - \frac{L_1}{2} \beta^2 \right) \sum_{k=k_0+1}^{T} \|\nabla_x F(x_k, t_k)\|^2 \leq F(x_{k_0+1}, t_{k_0+1}) - F(x_{T+1}, t_{T+1}) + L_0 \sqrt{d} \sum_{k=k_0+1}^{T} |t_{k+1} - t_k|$$

$$\leq F(x_{k_0+1}, t_{k_0+1}) - f^* + L_0 \sqrt{d} \sum_{k=k_0+1}^{T} |t_{k+1} - t_k|$$

$$\leq f(x_{k_0+1}) - f^* + L_0 \sqrt{d} \left( t_{k_0+1} + \sum_{k=k_0+1}^{T} |t_{k+1} - t_k| \right),$$

(10)

where the last inequality holds due to Lemma 3.3. Then, we can get the upper bound for $\|\nabla f(\hat{x})\|^2$ as

$$
\begin{aligned}
\|\nabla f(\hat{x})\|^2 &= \min_{k \in [T]} \|\nabla f(x_k)\|^2 \\
&\leq \min_{k=k_0+1,\dots,T} \|\nabla f(x_k)\|^2 \\
&\leq \frac{1}{T-k_0} \sum_{k=k_0+1}^{T} \|\nabla f(x_k)\|^2 \\
&\leq \frac{1}{T-k_0} \sum_{k=k_0+1}^{T} \|\nabla_x F(x_k, t_k)\|^2 + \frac{1}{T-k_0} L_0 L_1 (d+3)^{3/2} \sum_{k=k_0+1}^{T} t_k \\
&\leq \frac{2\left(f(x_{k_0+1}) - f^* + L_0\sqrt{d}\left(t_{k_0+1} + \sum_{k=k_0+1}^{T} |t_{k+1} - t_k|\right)\right)}{(T-k_0)(2\beta - L_1\beta^2)} + \frac{1}{T-k_0} L_0 L_1 (d+3)^{3/2} \sum_{k=k_0+1}^{T} t_k,
\end{aligned}
$$

where the third inequality holds due to Lemma B.2 (ii) and the last inequality follows from (10). If we choose the step size $\beta$ as $\frac{1}{L_1}$, we have

$$
\begin{aligned}
&\|\nabla f(\hat{x})\|^2 \\
&\leq \frac{2L_1\left(f(x_{k_0+1}) - f^* + L_0\sqrt{d}\left(t_{k_0+1} + \sum_{k=k_0+1}^{T} |t_{k+1} - t_k|\right)\right)}{T-k_0} + \frac{1}{T-k_0} L_0 L_1 (d+3)^{3/2} \sum_{k=k_0+1}^{T} t_k \\
&= O\left(\frac{1}{T-k_0}\left(1 + d^{3/2} \sum_{k=k_0+1}^{T} t_k\right)\right), \tag{11}
\end{aligned}
$$

where the last equality holds since $\sum_{k=k_0+1}^{T} |t_{k+1} - t_k| = O\left(\sum_{k=k_0+1}^{T} t_k\right)$ is satisfied. If we update $t_k$ as in Algorithm 2, we have $\sum_{k=k_0+1}^{T} t_k \leq \sum_{k=k_0+1}^{T} \max\{t_1 \gamma^{k-1}, \epsilon'\} \leq \sum_{k=k_0+1}^{T} \left(t_1 \gamma^{k-1} + \epsilon'\right) \leq \frac{t_1 \gamma^{k_0}}{1-\gamma} + \epsilon'(T - k_0)$. By taking $\epsilon'$ sufficiently close to 0, together with the assumption of $1/(1-\gamma) = O(1)$, we have $\sum_{k=k_0+1}^{T} t_k = O(\gamma^{k_0})$. This implies that $\|\nabla f(\hat{x})\|^2 \leq O(\frac{1+\gamma^{k_0} d^{3/2}}{T-k_0})$. Hence, we can obtain $\|\nabla f(\hat{x})\| \leq \epsilon$ in $T = k_0 + O\left(\frac{1+\gamma^{k_0} d^{3/2}}{\epsilon^2}\right)$ iterations.

Now, set $k_0$ as $k_0 = O\left(\frac{1}{\epsilon^2}\right)$, then, the iteration complexity can be bounded as $T = O\left(\frac{d^{3/2}}{\epsilon^2}\right)$. Furthermore, when $\gamma$ is chosen as $\gamma \leq d^{-3\epsilon^2/2}$, we can obtain $\gamma^{k_0} = O\left(d^{-3/2}\right)$ for some $k_0 = O\left(\frac{1}{\epsilon^2}\right)$. This yields the iteration complexity of $T = O\left(\frac{1}{\epsilon^2}\right)$.

$\square$

Before going to the proof of Theorem 3.5 in the stochastic setting, we prove that the gradient of the smoothed stochastic function $\nabla F(x, t; \xi)$ is unbiased, and it has a finite variance.

**Lemma B.3.** *Suppose that $f$ satisfies Assumption A1 (i) and Assumption A2.*
*(i) The stochastic gradient of the smoothed function $\nabla_x \bar{F}(x, t; \xi)$ becomes an unbiased estimator of $\nabla_x F(x, t)$. That is, for any $x \in \mathbb{R}^d$ and $t > 0$, $\mathbb{E}_\xi[\nabla_x \bar{F}(x, t; \xi)] = \nabla_x F(x, t)$ holds.*
*(ii) For any $x \in \mathbb{R}^d$ and $t > 0$, the variance of $\nabla_x \bar{F}(x, t; \xi)$ is bounded as $\mathbb{E}_\xi[\|\nabla_x \bar{F}(x, t; \xi) - \nabla_x F(x, t)\|^2] \leq \sigma^2$.*

**Proof for (i):** From Assumption A1 (i), we can exchange the order of integration in terms of $\xi$ and $u$, which yields that

$$
\begin{aligned}
\mathbb{E}_\xi[\nabla_x \bar{F}(x,t;\xi)] &= \mathbb{E}_\xi\left[\mathbb{E}_u\left[\frac{\bar{f}(x+tu;\xi)-\bar{f}(x;\xi)}{t}u\right]\right] \\
&= \mathbb{E}_u\left[\mathbb{E}_\xi\left[\frac{\bar{f}(x+tu;\xi)-\bar{f}(x;\xi)}{t}u\right]\right] \\
&= \mathbb{E}_u\left[\frac{f(x+tu)-f(x)}{t}u\right] \\
&= \nabla_x F(x,t).
\end{aligned}
$$

**Proof for (ii):** We have

$$
\begin{aligned}
\mathbb{E}_\xi[\|\nabla_x \bar{F}(x,t;\xi) - \nabla_x F(x,t)\|^2] &= \mathbb{E}_\xi[\|\nabla_x \mathbb{E}_u[\bar{f}(x+tu;\xi)] - \nabla_x \mathbb{E}_u[f(x+tu)]\|^2] \\
&= \mathbb{E}_\xi[\|\mathbb{E}_u[\nabla_x \bar{f}(x+tu;\xi) - \nabla f(x+tu)]\|^2] \\
&\leq \mathbb{E}_\xi[\mathbb{E}_u[\|\nabla_x \bar{f}(x+tu;\xi) - \nabla f(x+tu)\|^2]] \\
&= \mathbb{E}_u[\mathbb{E}_\xi[\|\nabla_x \bar{f}(x+tu;\xi) - \nabla f(x+tu)\|^2]] \\
&\leq \sigma^2,
\end{aligned}
$$

where the second and third equalities hold due to Assumption A1 (i), and the last inequality follows from Assumption A2 (ii).

**Proof for Theorem 3.5:** Denote $\delta_k := \nabla_x \bar{F}(x_k,t_k;\xi_k) - \nabla_x F(x_k,t_k)$. We follow the convergence analysis of stochastic gradient descent. According to Lemma 3.2, since $f(x)$ is $L_0$-*Lipschitz* and $L_1$-*smooth*, $F(x,t)$ is also $L_0$-*Lipschitz* and $L_1$-*smooth* in terms of $x$. Thus, we have

$$
\begin{aligned}
F(x_{k+1},t_k) &\leq F(x_k,t_k) + \langle \nabla_x F(x_k,t_k),(x_{k+1}-x_k)\rangle + \frac{L_1}{2}\|x_{k+1}-x_k\|^2 \\
&= F(x_k,t_k) - \beta\langle \nabla_x F(x_k,t_k),\nabla_x \bar{F}(x_k,t_k;\xi_k)\rangle + \frac{L_1}{2}\beta^2\|\nabla_x \bar{F}(x_k,t_k;\xi_k)\|^2 \\
&= F(x_k,t_k) - \left(\beta - \frac{L_1}{2}\beta^2\right)\|\nabla_x F(x_k,t_k)\|^2 - (\beta - L_1\beta^2)\langle \nabla_x F(x_k,t_k),\delta_k\rangle + \frac{L_1}{2}\beta^2\|\delta_k\|^2,
\end{aligned}
\tag{12}
$$

where the first equation holds due to the updating rule $x_{k+1} - x_k = -\beta\nabla_x \bar{F}(x_k,t_k;\xi_k)$, and the last equation holds due to the definition of $\delta_k$. Denote

$$
A_k := -(\beta - L_1\beta^2)\langle \nabla_x F(x_k,t_k),\delta_k\rangle + \frac{L_1}{2}\beta^2\|\delta_k\|^2
$$

for simplicity. From (12), we obtain the upper bound for $\|\nabla_x F(x,t)\|^2$ as follows:

$$
\begin{aligned}
\left(\beta - \frac{L_1}{2}\beta^2\right)\|\nabla_x F(x_k,t_k)\|^2 &\leq F(x_k,t_k) - F(x_{k+1},t_k) + A_k \\
&= F(x_k,t_k) - F(x_{k+1},t_{k+1}) + F(x_{k+1},t_{k+1}) - F(x_{k+1},t_k) + A_k \\
&\leq F(x_k,t_k) - F(x_{k+1},t_{k+1}) + L_0|t_{k+1}-t_k|\sqrt{d} + A_k,
\end{aligned}
$$

where the last inequality follows from Lemma 3.3.

Now, sum up the above inequality for all iterations $k_0 + 1 \le k \le T$ ($k_0 < T$). Then we have

$$\left(\beta - \frac{L_1}{2}\beta^2\right) \sum_{k=k_0+1}^{T} \|\nabla_x F(x_k, t_k)\|^2$$

$$\le F(x_{k_0+1}, t_{k_0+1}) - F(x_{T+1}, t_{T+1}) + L_0\sqrt{d} \sum_{k=k_0+1}^{T} |t_{k+1} - t_k| + \sum_{k=k_0+1}^{T} A_k$$

$$\le F(x_{k_0+1}, t_{k_0+1}) - f^* + L_0\sqrt{d} \sum_{k=k_0+1}^{T} |t_{k+1} - t_k| + \sum_{k=k_0+1}^{T} A_k.$$

$$\le f(x_{k_0+1}) - f^* + L_0\sqrt{d}\left(t_{k_0+1} + \sum_{k=k_0+1}^{T} |t_{k+1} - t_k|\right) + \sum_{k=k_0+1}^{T} A_k.$$

Take the expectation with respect to the random vectors $\{\xi_{k_0+1}, \ldots, \xi_T\}$, then we have

$$\left(\beta - \frac{L_1}{2}\beta^2\right) \sum_{k=k_0+1}^{T} \mathbb{E}_\xi[\|\nabla_x F(x_k, t_k)\|^2]$$

$$\le f(x_{k_0+1}) - f^* + L_0\sqrt{d}\left(t_{k_0+1} + \sum_{k=k_0+1}^{T} \mathbb{E}_\xi[|t_{k+1} - t_k|]\right) + \sum_{k=k_0+1}^{T} \mathbb{E}_\xi[A_k]. \quad (13)$$

The expectation of $A_k$ is evaluated as

$$\sum_{k=k_0+1}^{T} \mathbb{E}_\xi[A_k] = - \sum_{k=k_0+1}^{T} (\beta - L_1\beta^2)\mathbb{E}_\xi[\langle \nabla_x F(x_k, t_k), \delta_k \rangle] + \sum_{k=k_0+1}^{T} \frac{L_1}{2}\beta^2 \mathbb{E}_\xi[\|\delta_k\|^2]$$

$$\le (T - k_0)\frac{L_1}{2}\beta^2\sigma^2, \quad (14)$$

where the last equality holds due to Lemma B.3 (ii) ($\mathbb{E}_\xi[\|\delta_k\|^2] \le \sigma^2$) and the fact that each point $x_k$ is a function of the history $\xi_{[k-1]}$ in the random process, thus $\mathbb{E}_{\xi_k}[\langle \nabla_x F(x_k, t_k), \delta_k \rangle \mid \xi_{[k-1]}] = 0$.

Then, we can estimate the upper bound for $\mathbb{E}_{\xi,k'}[\|\nabla f(\hat{x})\|^2]$ as

$$\mathbb{E}_{\xi,k'}[\|\nabla f(\hat{x})\|^2] = \frac{1}{T - k_0} \sum_{k=k_0+1}^{T} \mathbb{E}_\xi[\|\nabla f(x_k)\|^2]$$

$$\le \frac{1}{T - k_0} \sum_{k=k_0+1}^{T} \mathbb{E}_\xi[\|\nabla_x F(x_k, t_k)\|^2] + \frac{1}{T - k_0}L_0 L_1(d + 3)^{3/2} \sum_{k=k_0+1}^{T} \mathbb{E}_\xi[t_k]$$

$$\le \frac{2\left(f(x_{k_0+1}) - f^* + L_0\sqrt{d}\left(t_{k_0+1} + \sum_{k=k_0+1}^{T} \mathbb{E}_\xi[|t_{k+1} - t_k|]\right)\right)}{(T - k_0)(2\beta - L_1\beta^2)}$$

$$+ \frac{1}{T - k_0}L_0 L_1(d + 3)^{3/2} \sum_{k=k_0+1}^{T} \mathbb{E}_\xi[t_k] + \frac{L_1\beta^2\sigma^2}{2\beta - L_1\beta^2},$$

where the first inequality holds due to Lemma B.2 (ii) and the last inequality follows from (13) and (14).

If the step size $\beta$ is chosen as $\beta = \min\{\frac{1}{L_1}, \frac{1}{\sqrt{T-k_0}}\}$, then we have

$$\frac{1}{2\beta - L_1\beta^2} \le \frac{1}{\beta},$$

$$\frac{1}{\beta} \le L_1 + \sqrt{T - k_0}.$$

Hence, we can obtain

$$\frac{2\left(f(x_{k_0+1}) - f^* + L_0\sqrt{d}\left(t_{k_0+1} + \sum_{k=k_0+1}^T \mathbb{E}_\xi[|t_{k+1} - t_k|]\right)\right)}{(T-k_0)(2\beta - L_1\beta^2)}$$

$$+ \frac{1}{T-k_0}L_0L_1(d+3)^{3/2}\sum_{k=k_0+1}^T \mathbb{E}_\xi[t_k] + \frac{L_1\beta^2\sigma^2}{2\beta - L_1\beta^2}$$

$$= O\left(\frac{1 + \sqrt{d}\mathbb{E}_\xi\left[\sum_{k=k_0+1}^T |t_{k+1} - t_k|\right]}{\sqrt{T-k_0}} + \frac{d^{3/2}}{T-k_0}\mathbb{E}_\xi\left[\sum_{k=k_0+1}^T t_k\right]\right).$$

If $t_k$ is updated as in Algorithm 2, we have $\sum_{k=k_0+1}^T |t_{k+1} - t_k| \leq t_1\gamma^{k_0} = O(\gamma^{k_0})$ and $\sum_{k=k_0+1}^T t_k \leq \frac{t_1\gamma^{k_0}}{1-\gamma} + \epsilon'T = O(\gamma^{k_0})$ in the same argument that showed Theorem 3.4. Combining the above inequalities, we obtain

$$\mathbb{E}_{\xi,k'}[\|\nabla f(\hat{x})\|^2] = \frac{1}{T-k_0}\sum_{k=k_0+1}^T \mathbb{E}_\xi[\|\nabla f(x_k)\|^2] = O\left(\frac{1 + \sqrt{d}\gamma^{k_0}}{\sqrt{T-k_0}} + \frac{d^{3/2}\gamma^{k_0}}{T-k_0}\right). \quad (15)$$

Here, we have $k_0 = O\left(\frac{1}{\epsilon^4}\right)$ by the definition of $k_0$. Thus, by setting $T = k_0 + O\left(\frac{d}{\epsilon^4} + \frac{d^{3/2}}{\epsilon^2}\right) = O\left(\frac{d}{\epsilon^4} + \frac{d^{3/2}}{\epsilon^2}\right)$, we can obtain $\mathbb{E}_{\xi,k'}[\|\nabla f(\hat{x})\|^2] \leq \epsilon^2$. This implies $\mathbb{E}_{\xi,k'}[\|\nabla f(\hat{x})\|] \leq \epsilon$ as $\mathbb{E}_{\xi,k'}[\|\nabla f(\hat{x})\|]^2 \leq \mathbb{E}_{\xi,k'}[\|\nabla f(\hat{x})\|^2]$ follows from Jensen's inequality. Furthermore, when $\gamma$ is chosen as $\gamma \leq (\max\{d^{1/2}, d^{3/2}\epsilon^2\})^{-\epsilon^4}$, we have $\log_\gamma \min\{d^{-1/2}, d^{-3/2}\epsilon^{-2}\} = O\left(\frac{1}{\epsilon^4}\right)$, which implies $k_0 = \Omega(\log_\gamma \min\{d^{-1/2}, d^{-3/2}\epsilon^{-2}\})$. Therefore, we can obtain $\gamma^{k_0} = O\left(\min\{d^{-1/2}, d^{-3/2}\epsilon^{-2}\}\right)$, which yields the iteration complexity of $T = O\left(\frac{1}{\epsilon^4}\right)$.

$\square$

### B.3 Zeroth-order SLGH algorithm

In the zeroth-order setting, we can evaluate the gap between the zeroth-order gradient estimator and the true gradient using the following lemma.

**Lemma B.4 (Theorem 4 in [28]).** *Let $f$ be a $L_1$-smooth function, then for any $x \in \mathbb{R}^d$ and for any $t > 0$, we have*

$$\mathbb{E}_u\left[\frac{1}{t^2}(f(x+tu) - f(x))^2\|u\|^2\right] \leq \frac{t^2}{2}L_1^2(d+6)^3 + 2(d+4)\|\nabla f(x)\|^2.$$

**Proof for Theorem 4.1:** Let $w_k := (u_k, v_k)$, $k \in [T]$, and denote $\delta_k := \tilde{g}_x(x_k, t_k; u_k) - \nabla_x F(x_k, t_k)$, where $\tilde{g}_x(x_k, t_k; u_k)$ is the zeroth-order estimator of gradient defined in the main paper. Utilize the updating rule of $x$ and $L_1$-smoothness of $F(x,t)$ in terms of $x$. Then we have

$$F(x_{k+1}, t_k) \leq F(x_k, t_k) + \langle \nabla_x F(x_k, t_k), (x_{k+1} - x_k)\rangle + \frac{L_1}{2}\|x_{k+1} - x_k\|^2$$

$$= F(x_k, t_k) - \beta\langle \nabla_x F(x_k, t_k), \tilde{g}_x(x_k, t_k; u_k)\rangle + \frac{L_1}{2}\beta^2\|\tilde{g}_x(x_k, t_k; u_k)\|^2$$

$$= F(x_k, t_k) - \beta\|\nabla_x F(x_k, t_k)\|^2 - \beta\langle \nabla_x F(x_k, t_k), \delta_k\rangle + \frac{L_1}{2}\beta^2\|\tilde{g}_x(x_k, t_k; u_k))\|^2, \quad (16)$$

where the first equation holds due to the updating rule $x_{k+1} - x_k = -\beta\tilde{g}_x(x_k, t_k; u_k)$.
Denote

$$B_k := -\beta\langle \nabla_x F(x_k, t_k), \delta_k\rangle + \frac{L_1}{2}\beta^2\|\tilde{g}_x(x_k, t_k; u_k)\|^2$$

for simplicity. From Lemma 3.3 and (16), we get the upper bound for $\|\nabla_x F(x,t)\|^2$ as

$$\beta \|\nabla_x F(x_k, t_k)\|^2 \leq F(x_k, t_k) - F(x_{k+1}, t_k) + B_k$$
$$= F(x_k, t_k) - F(x_{k+1}, t_{k+1}) + F(x_{k+1}, t_{k+1}) - F(x_{k+1}, t_k) + B_k$$
$$\leq F(x_k, t_k) - F(x_{k+1}, t_{k+1}) + L_0 |t_{k+1} - t_k|\sqrt{d} + B_k.$$

Now, sum up the above inequality for all iterations $k_0 + 1 \leq k \leq T$ ($k_0 < T$). Then we have

$$\sum_{k=k_0+1}^{T} \beta \|\nabla_x F(x_k, t_k)\|^2 \leq F(x_{k_0+1}, t_{k_0+1}) - F(x_{T+1}, t_{T+1}) + L_0 \sum_{k=k_0+1}^{T} |t_{k+1} - t_k|\sqrt{d} + \sum_{k=k_0+1}^{T} B_k$$

$$\leq F(x_{k_0+1}, t_{k_0+1}) - f^* + L_0\sqrt{d} \sum_{k=k_0+1}^{T} |t_{k+1} - t_k| + \sum_{k=k_0+1}^{T} B_k$$

$$\leq f(x_{k_0+1}) - f^* + L_0\sqrt{d} \left( t_{k_0+1} + \sum_{k=k_0+1}^{T} |t_{k+1} - t_k| \right) + \sum_{k=k_0+1}^{T} B_k.$$

Next, take the expectations with respect to random vectors $\{w_{k_0+1}, \ldots, w_T\}$ on both sides. Then we can get

$$\sum_{k=k_0+1}^{T} \beta \mathbb{E}_w[\|\nabla_x F(x_k, t_k)\|^2] \leq f(x_{k_0+1}) - f^* + L_0\sqrt{d} \left( t_{k_0+1} + \sum_{k=k_0+1}^{T} \mathbb{E}_w[|t_{k+1} - t_k|] \right)$$

$$+ \sum_{k=k_0+1}^{T} \mathbb{E}_w[B_k]. \tag{17}$$

Observe by the definition of $\tilde{g}_x(x_k, t_k; u_k)$ in the main paper that $\mathbb{E}_{u_k}[\tilde{g}_x(x_k, t_k; u_k) \mid u_{[k-1]}] = \nabla_x F(x_k, t_k)$, thus $\mathbb{E}_{w_k}[\langle \nabla_x F(x_k, t_k), \delta_k \rangle \mid w_{[k-1]}] = 0$ holds. Then we have

$$\mathbb{E}_{w_k}[B_k \mid w_{[k-1]}] = -\beta \mathbb{E}_{w_k}[\langle \nabla_x F(x_k, t_k), \delta_k \rangle \mid w_{[k-1]}] + \frac{L_1}{2}\beta^2 \mathbb{E}_{w_k}[\|\tilde{g}_x(x_k, t_k; u_k)\|^2 \mid w_{[k-1]}]$$

$$\leq \frac{L_1}{2}\beta^2 \left( \frac{\mathbb{E}_{w_k}[t_k^2 \mid w_{[k-1]}]}{2} L_1^2 (d+6)^3 + 2(d+4)\mathbb{E}_{w_k}[\|\nabla f(x_k)\|^2 \mid w_{[k-1]}] \right)$$

$$= \frac{\mathbb{E}_{w_k}[t_k^2 \mid w_{[k-1]}]}{4} L_1^3 \beta^2 (d+6)^3 + L_1 \beta^2 (d+4)\mathbb{E}_{w_k}[\|\nabla f(x_k)\|^2 \mid w_{[k-1]}], \tag{18}$$

where the inequality holds due to Lemma B.4.

Lemma B.2 (ii) together with the above inequalities yields that

$$\sum_{k=k_0+1}^{T} \beta \mathbb{E}_w[\|\nabla f(x_k)\|^2]$$

$$\leq \sum_{k=k_0+1}^{T} \beta \mathbb{E}_w[\|\nabla_x F(x_k, t_k)\|^2] + \sum_{k=k_0+1}^{T} \beta L_0 L_1 (d+3)^{3/2} \mathbb{E}_w[t_k]$$

$$\leq f(x_{k_0+1}) - f^* + L_0\sqrt{d} \left( t_{k_0+1} + \sum_{k=k_0+1}^{T} \mathbb{E}_w[|t_{k+1} - t_k|] \right) + \sum_{k=k_0+1}^{T} \mathbb{E}_w[B_k]$$

$$+ \sum_{k=k_0+1}^{T} \beta L_0 L_1 (d+3)^{3/2} \mathbb{E}_w[t_k]$$

$$\leq f(x_{k_0+1}) - f^* + L_0\sqrt{d} \left( t_{k_0+1} + \sum_{k=k_0+1}^{T} \mathbb{E}_w[|t_{k+1} - t_k|] \right) + \sum_{k=k_0+1}^{T} \frac{\mathbb{E}_w[t_k^2]}{4} L_1^3 \beta^2 (d+6)^3$$

$$+ \sum_{k=k_0+1}^{T} L_1 \beta^2 (d+4)\mathbb{E}_w[\|\nabla f(x_k)\|^2] + \sum_{k=k_0+1}^{T} \beta L_0 L_1 (d+3)^{3/2} \mathbb{E}_w[t_k], \tag{19}$$

where the second inequality holds due to (17), and the last inequality follows from (18). Rearrange the terms in the above inequality. Then we can get

$$(\beta - (d+4)L_1\beta^2) \sum_{k=k_0+1}^{T} \mathbb{E}_w[\|\nabla f(x_k)\|^2] \leq f(x_{k_0+1}) - f^* + L_0\sqrt{d}\left(t_{k_0+1} + \sum_{k=k_0+1}^{T} \mathbb{E}_w[|t_{k+1} - t_k|]\right)$$
$$+ \frac{L_1^3\beta^2(d+6)^3}{4} \sum_{k=k_0+1}^{T} \mathbb{E}_w[t_k^2] + L_0L_1\beta(d+3)^{3/2} \sum_{k=k_0+1}^{T} \mathbb{E}_w[t_k].$$
(20)

Divide both sides of the above inequality by $(T - k_0)(\beta - (d+4)L_1\beta^2)$ and set the step size $\beta$ as $\frac{1}{2(d+4)L_1}$. Since $\frac{1}{\beta-(d+4)L_1\beta^2} \leq 4(d+4)L_1$ holds, we can obtain

$$\frac{1}{T-k_0} \sum_{k=k_0+1}^{T} \mathbb{E}_w[\|\nabla f(x_k)\|^2] \leq \frac{4(d+4)L_1}{T-k_0}\left(f(x_{k_0+1}) - f^* + L_0\sqrt{d}\left(t_{k_0+1} + \sum_{k=k_0+1}^{T} \mathbb{E}_w[|t_{k+1} - t_k|]\right)\right.$$
$$\left. + \frac{L_1(d+6)^3}{16(d+4)^2} \sum_{k=k_0+1}^{T} \mathbb{E}_w[t_k^2] + \frac{L_0(d+3)^{3/2}}{2(d+4)} \sum_{k=k_0+1}^{T} \mathbb{E}_w[t_k]\right)$$
$$= O\left(\frac{d}{T-k_0}\left(1 + d\mathbb{E}_w\left[\sum_{k=k_0+1}^{T} t_k^2\right] + \sqrt{d}\mathbb{E}_w\left[\sum_{k=k_0+1}^{T} t_k\right]\right)\right)$$
$$= O\left(\frac{d}{T-k_0}\left(1 + d\gamma^{2k_0} + \sqrt{d}\gamma^{k_0}\right)\right),$$
(21)

where the last equality follows from the update rule of $t_k$, as shown in the proof of Theorem 3.4 as well.

Here, we have $k_0 = O\left(\frac{d}{\epsilon^2}\right)$ by the definition of $k_0$. Thus, by setting $T = k_0 + O\left(\frac{d^2}{\epsilon^2}\right) = O\left(\frac{d^2}{\epsilon^2}\right)$, we can obtain $\mathbb{E}_{w,k'}[\|\nabla f(\hat{x})\|^2] = \frac{1}{T-k_0}\sum_{k=k_0+1}^{T} \mathbb{E}_w[\|\nabla f(x_k)\|^2] \leq \epsilon^2$. This implies $\mathbb{E}_{w,k'}[\|\nabla f(\hat{x})\|] \leq \epsilon$ as $\mathbb{E}_{w,k'}[\|\nabla f(\hat{x})\|]^2 \leq \mathbb{E}_{w,k'}[\|\nabla f(\hat{x})\|^2]$ follows from Jensen's inequality. Furthermore, when $\gamma$ is chosen as $\gamma \leq d^{-\epsilon^2/2d}$, we have $\log_\gamma d^{-1/2} = O\left(\frac{d}{\epsilon^2}\right)$, which implies $k_0 = \Omega\left(\log_\gamma d^{-1/2}\right)$. Therefore, we can obtain $\gamma^{k_0} = O(d^{-1/2})$, which yields the iteration complexity of $T = O\left(\frac{d}{\epsilon^2}\right)$.

$\square$

**Proof for Theorem 4.2:** Let $\zeta_k := (\xi_k, u_k, v_k)$, $k \in [T]$ and denote $\delta_k := \tilde{G}_x(x_k, t_k; \xi_k, u_k) - \nabla_x F(x_k, t_k)$. As discussed in the main paper, we have

$$\mathbb{E}_{\xi,u}[\tilde{G}_x(x, t; \xi, u)] = \mathbb{E}_u[\mathbb{E}_\xi[\tilde{G}_x(x, t; \xi, u)|u]] = \nabla_x F(x, t).$$
(22)

From the update rule for $x$, we can obtain

$$F(x_{k+1}, t_k) \leq F(x_k, t_k) + \langle \nabla_x F(x_k, t_k), (x_{k+1} - x_k)\rangle + \frac{L_1}{2}\|x_{k+1} - x_k\|^2$$
$$= F(x_k, t_k) - \beta\langle \nabla_x F(x_k, t_k), \tilde{G}_x(x_k, t_k; \xi_k, u_k)\rangle + \frac{L_1}{2}\beta^2\|\tilde{G}_x(x_k, t_k; \xi_k, u_k)\|^2$$
$$= F(x_k, t_k) - \beta\|\nabla_x F(x_k, t_k)\|^2 - \beta\langle \nabla_x F(x_k, t_k), \delta_k\rangle + \frac{L_1}{2}\beta^2\|\tilde{G}_x(x_k, t_k; \xi_k, u_k)\|^2.$$

Now, denote

$$D_k := -\beta\langle \nabla_x F(x_k, t_k), \delta_k\rangle + \frac{L_1}{2}\beta^2\|\tilde{G}_x(x_k, t_k; \xi_k, u_k)\|^2$$

for simplicity. Then, we can get the upper bound for $\|\nabla_x F(x, t)\|^2$ with $D_k$:

$$\beta\|\nabla_x F(x_k, t_k)\|^2 \leq F(x_k, t_k) - F(x_{k+1}, t_k) + D_k$$
$$= F(x_k, t_k) - F(x_{k+1}, t_{k+1}) + F(x_{k+1}, t_{k+1}) - F(x_{k+1}, t_k) + D_k$$
$$\leq F(x_k, t_k) - F(x_{k+1}, t_{k+1}) + L_0|t_{k+1} - t_k|\sqrt{d} + D_k.$$

Sum up the above inequality for all iterations $k_0 + 1 \leq k \leq T$ ($T > k_0$). Then we have

$$\sum_{k=k_0+1}^{T} \beta \|\nabla_x F(x_k, t_k)\|^2$$

$$\leq F(x_{k_0+1}, t_{k_0+1}) - F(x_{T+1}, t_{T+1}) + L_0\sqrt{d} \sum_{k=k_0+1}^{T} |t_{k+1} - t_k| + \sum_{k=k_0+1}^{T} D_k$$

$$\leq F(x_{k_0+1}, t_{k_0+1}) - f^* + L_0\sqrt{d} \sum_{k=k_0+1}^{T} |t_{k+1} - t_k| + \sum_{k=k_0+1}^{T} D_k$$

$$\leq f(x_{k_0+1}) - f^* + L_0\sqrt{d} \left( t_{k_0+1} + \sum_{k=k_0+1}^{T} |t_{k+1} - t_k| \right) + \sum_{k=k_0+1}^{T} D_k, \tag{23}$$

where the last inequality follows from Lemma 3.3. Observe from (22) that

$$\mathbb{E}_{\zeta_k} [\langle \nabla_x F(x_k, t_k), \delta_k \rangle \mid \zeta_{[k-1]}] = 0.$$

Thus, we have

$$\mathbb{E}_{\zeta_k}[D_k \mid \zeta_{[k-1]}] = -\beta \mathbb{E}_{\zeta_k}[\langle \nabla_x F(x_k, t_k), \delta_k \rangle \mid \zeta_{[k-1]}] + \frac{L_1}{2}\beta^2 \mathbb{E}_{\zeta_k}[\|\tilde{G}_x(x_k, t_k; \xi_k, u_k)\|^2 \mid \zeta_{[k-1]}]$$

$$= \frac{L_1}{2}\beta^2 \mathbb{E}_{\zeta_k}(\|\tilde{G}_x(x_k, t_k; \xi_k, u_k)\|^2 \mid \zeta_{[k-1]})$$

$$\leq \frac{L_1}{2}\beta^2 \left( \frac{\mathbb{E}_{\zeta_k}[t_k^2 \mid \zeta_{[k-1]}]}{2} L_1^2(d+6)^3 + 2(d+4)(\mathbb{E}_{\zeta_k}[\|\nabla_x \bar{f}(x_k; \xi_k) \mid \zeta_{[k-1]}\|^2]) \right)$$

$$\leq \frac{L_1}{2}\beta^2 \left( \frac{\mathbb{E}_{\zeta_k}[t_k^2 \mid \zeta_{[k-1]}]}{2} L_1^2(d+6)^3 + 2(d+4)(\mathbb{E}_{\zeta_k}[\|\nabla f(x_k) \mid \zeta_{[k-1]}\|^2] + \sigma^2) \right), \tag{24}$$

where the fist inequality follows from Lemma B.4 and the last inequality holds due to Assumption A2 (ii).

Take the expectation for (23) with respect to $\zeta_{k_0+1}, \ldots, \zeta_T$. Together with Lemma B.2 (ii), we have

$$\sum_{k=k_0+1}^{T} \beta \mathbb{E}_\zeta[\|\nabla f(x_k)\|^2]$$

$$\leq \sum_{k=k_0+1}^{T} \beta \mathbb{E}_\zeta[\|\nabla_x F(x_k, t_k)\|^2] + \sum_{k=k_0+1}^{T} \beta \mathbb{E}_\zeta[t_k] L_0 L_1 (d+3)^{3/2}$$

$$\leq f(x_{k_0+1}) - f^* + L_0\sqrt{d} \left( t_{k_0+1} + \sum_{k=k_0+1}^{T} \mathbb{E}_\zeta[|t_{k+1} - t_k|] \right) + \sum_{k=k_0+1}^{T} \mathbb{E}_\zeta[D_k]$$

$$+ \sum_{k=k_0+1}^{T} \mathbb{E}_\zeta[t_k] L_0 L_1 \beta (d+3)^{3/2}$$

$$\leq f(x_{k_0+1}) - f^* + L_0\sqrt{d} \left( t_{k_0+1} + \sum_{k=k_0+1}^{T} \mathbb{E}_\zeta[|t_{k+1} - t_k|] \right) + \sum_{k=k_0+1}^{T} \mathbb{E}_\zeta[t_k] L_0 L_1 \beta (d+3)^{3/2}$$

$$+ \sum_{k=k_0+1}^{T} \frac{\mathbb{E}_\zeta[t_k^2]}{4} L_1^3 \beta^2 (d+6)^3 + \sum_{k=k_0+1}^{T} L_1 \beta^2 (d+4) \mathbb{E}_\zeta[\|\nabla f(x_k)\|^2] + L_1 \beta^2 (d+4)\sigma^2 (T - k_0),$$

where the last inequality holds due to (24). Rearrange the terms in the above inequality. Then we can get

$$(\beta - (d+4)L_1\beta^2) \sum_{k=k_0+1}^{T} \mathbb{E}_\zeta[\|\nabla f(x_k)\|^2] \leq f(x_{k_0+1}) - f^* + L_0\sqrt{d}\left(t_{k_0+1} + \sum_{k=k_0+1}^{T} \mathbb{E}_\zeta[|t_{k+1} - t_k|]\right)$$

$$+ \frac{L_1^3\beta^2(d+6)^3}{4} \sum_{k=k_0+1}^{T} \mathbb{E}_\zeta[t_k^2] + L_1\beta^2(d+4)\sigma^2(T-k_0)$$

$$+ \sum_{k=k_0+1}^{T} \mathbb{E}_\zeta[t_k]L_0L_1\beta(d+3)^{3/2}, \qquad (25)$$

If the step size $\beta$ is chosen as $\min\left\{\frac{1}{2(d+4)L_1}, \frac{1}{\sqrt{(T-k_0)(d+4)}}\right\}$, then we have

$$\frac{1}{\beta - (d+4)L_1\beta^2} \leq \frac{2}{\beta}, \quad \frac{1}{\beta} \leq 2(d+4)L_1 + \sqrt{(T-k_0)(d+4)}.$$

Hence, by dividing both sides of (25) by $(T-k_0)(\beta - 2(d+4)L_1\beta^2)$, we can obtain

$$\frac{1}{T-k_0} \sum_{k=1}^{T} \mathbb{E}_\zeta[\|\nabla f(x_k)\|^2]$$

$$\leq \frac{f(x_{k_0+1}) - f^* + L_0\sqrt{d}\left(t_{k_0+1} + \sum_{k=k_0+1}^{T} \mathbb{E}_\zeta[|t_{k+1} - t_k|]\right) + L_0L_1(d+3)^{3/2}\beta \sum_{k=k_0+1}^{T} \mathbb{E}_\zeta[t_k]}{(T-k_0)(\beta - (d+4)L_1\beta^2)}$$

$$+ \frac{\frac{L_1^3\beta^2(d+6)^3}{4} \sum_{k=k_0+1}^{T} \mathbb{E}_\zeta[t_k^2] + L_1\beta^2(d+4)\sigma^2 T}{(T-k_0)(\beta - (d+4)L_1\beta^2)}$$

$$\leq \frac{2}{T-k_0}\left(f(x_{k_0+1}) - f^* + L_0\sqrt{d}\left(t_{k_0+1} + \sum_{k=k_0+1}^{T} \mathbb{E}_\zeta[|t_{k+1} - t_k|]\right)\right)\left(2(d+4)L_1 + \sqrt{(T-k_0)(d+4)}\right)$$

$$+ \frac{2}{T-k_0}L_0L_1(d+3)^{3/2} \sum_{k=k_0+1}^{T} \mathbb{E}_\zeta[t_k] + \frac{L_1^3\beta(d+6)^3}{2(T-k_0)} \sum_{k=k_0+1}^{T} \mathbb{E}_\zeta[t_k^2] + 2L_1\beta(d+4)\sigma^2$$

$$= O\left(\frac{\sqrt{d}\left(1 + \sqrt{d}\sum_{k=k_0+1}^{T} \mathbb{E}_\zeta[|t_{k+1} - t_k|]\right)}{\sqrt{T-k_0}} + \frac{d\left(d\mathbb{E}_\zeta\left[\sum_{k=k_0+1}^{T} t_k^2\right] + \sqrt{d}\mathbb{E}_\zeta\left[\sum_{k=k_0+1}^{T} t_k\right] + 1\right)}{T-k_0}\right)$$

$$= O\left(\frac{\sqrt{d}\left(1 + \sqrt{d}\gamma^{k_0}\right)}{\sqrt{T-k_0}} + \frac{d\left(d\gamma^{2k_0} + \sqrt{d}\gamma^{k_0} + 1\right)}{T-k_0}\right)$$

where the last equality follows from the update rule of $t_k$, as shown in the proof of Theorem 3.4 as well.

Here, we have $k_0 = O\left(\frac{d}{\epsilon^4}\right)$ by the definition of $k_0$. Thus, by setting $T = k_0 + O\left(\frac{d^2}{\epsilon^4}\right) = O\left(\frac{d^2}{\epsilon^4}\right)$, we can obtain $\mathbb{E}_{\zeta,k'}[\|\nabla f(\hat{x})\|^2] = \frac{1}{T-k_0}\sum_{k=k_0+1}^{T} \mathbb{E}_\zeta[\|\nabla f(x_k)\|^2] \leq \epsilon^2$. This implies $\mathbb{E}_{\zeta,k'}[\|\nabla f(\hat{x})\|] \leq \epsilon$ as $\mathbb{E}_{\zeta,k'}[\|\nabla f(\hat{x})\|]^2 \leq \mathbb{E}_{\zeta,k'}[\|\nabla f(\hat{x})\|^2]$ follows from Jensen's inequality. Furthermore, when $\gamma$ is chosen as $\gamma \leq d^{-\epsilon^4/2d}$, we have $\log_\gamma d^{-1/2} = O\left(\frac{d}{\epsilon^4}\right)$, which implies that $k_0 = \Omega\left(\log_\gamma d^{-1/2}\right)$. Therefore, we can obtain $\gamma^{k_0} = O(d^{-1/2})$, which yields the iteration complexity of $T = O\left(\frac{d}{\epsilon^4}\right)$.

$\square$

## C ZOSLGH algorithm with error tolerance

In Sections 3 and 4, we assumed that we had access to the exact function value or a gradient oracle whose variance was finite. However, in some practical cases, we will have access only to the function values containing error, and it would be impossible to obtain accurate gradient oracles of an underlying objective function. Figure 2 illustrates such a case; although the objective function $f$ (Figure 2(a)) is smooth, the accessible function $f'$ (Figure 2(b)) contains some error, and thus many local minima arise. In this section, we consider optimizing a smooth objective function $f$ using only the information of $f'$. We assume that the following condition holds between $f$ and $f'$.

**Assumption A3.** The supremum norm of the difference between $f$ and $f'$ is uniformly bounded:

$$\sup_{x \in \mathbb{R}^d} |f(x) - f'(x)| \leq \nu.$$

In the stochastic setting, we assume $\sup_{x \in \mathbb{R}^d} |f(x; \xi) - f'(x; \xi)| \leq \nu$ for any $\xi$.

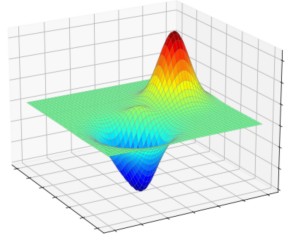

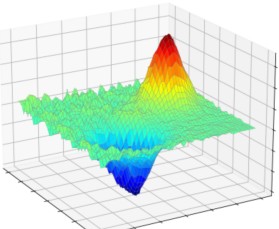

(a) Smooth objective function                    (b) Accessible function with error

Figure 2: Illustration of a smooth objective function and the accessible function that contains error.

Please note that we do not impose any other assumptions on the accessible function $f'$. Thus, $f'$ can be non-Lipschitz or even discontinuous. Even in such cases, we can develop an algorithm with a convergence guarantee because its smoothed function $F'(x, t)$ is smooth as far as $t$ is sufficiently large. In the following, we denote the Lipschitz and gradient Lipschitz constant of $F'(\cdot, t)$ as $L_0(t)$ and $L_1(t)$, respectively.

The ZOSLGH algorithm in this setting is almost the same as Algorithm 3. The only difference is $\sqrt{\nu}$ rather than $\epsilon$ in the update rule of $t_{k+1}$. See the Algorithm 4 for a more detailed description. Please note that $F', \tilde{g}'_x, \tilde{G}'_{x,u}, \tilde{g}'_t, \tilde{G}'_{t,v}$ are defined in the same way as the no-error setting using $f'$.

---

**Algorithm 4** Deterministic/Stochastic Zeroth-Order Single Loop GH algorithm (ZOSLGH) with error tolerance

---

**Require:** Iteration number $T$, initial solution $x_1$, initial smoothing parameter $t_1$, sequence of step sizes $\{\beta_k\}$ for $x$, step size $\eta$ for $t$, decreasing factor $\gamma \in (0, 1)$, error tolerance $\nu$

**for** $k = 1$ to $T$ **do**

Sample $u_k$ from $\mathcal{N}(0, \mathrm{I}_d)$

Update $x_k$ by

$$x_{k+1} = x_k - \beta_k \bar{G}'_{x,u},$$

$$\text{where } \bar{G}'_{x,u} = \begin{cases} \tilde{g}'_x(x_k, t_k; u_k) & \text{(deterministic)} \\ \tilde{G}'_x(x_k, t_k; \xi_k, u_k), \ \xi_k \sim P & \text{(stochastic)} \end{cases}$$

Sample $v_k$ from $\mathcal{N}(0, \mathrm{I}_d)$

Update $t_k$ by

$$t_{k+1} = \begin{cases} \max\{\gamma t_k, \sqrt{\nu}\} & \text{(SLGH}_r) \\ \max\{\min\{t_k - \eta \bar{G}'_{t,v}, \gamma t_k\}, \sqrt{\nu}\} & \text{(SLGH}_d) \end{cases},$$

$$\text{where } \bar{G}'_{t,v} = \begin{cases} \tilde{g}'_t(x_k, t_k; v_k) & \text{(deterministic)} \\ \tilde{G}'_t(x_k, t_k; \xi_k, v_k), \ \xi_k \sim P & \text{(stochastic)} \end{cases}$$

**end for**

---

We provide the convergence analyses in the following theorems. The definitions of $\hat{x}$ in the deterministic and stochastic settings are given in Appendix C.2 and C.3, respectively.

**Theorem C.1** (**Convergence of ZOSLGH with error tolerance, Deterministic setting**). *Suppose Assumptions A1 and A3 hold.*

*Take $k_1 := \Theta(d/\epsilon^2)$ and $k_2 := O\left(\log_\gamma 1/d\right)$ and define $k_0 = \min\{k_1, k_2\}$. Let $\hat{x} := x_{k'}$, where $k'$ is chosen from a uniform distribution over $\{k_0 + 1, k_0 + 2, \ldots, T\}$. Set the stepsize for $x$ at iteration $k$ as $\beta_k = \frac{1}{16(d+4)L_1(t_k)}$, $k \in [T]$. Then, for any setting of the parameter $\gamma$, if the error level $\nu$ satisfies $\nu = O(\epsilon^2/d^3)$, $\hat{x}$ satisfies $\mathbb{E}[\|\nabla f(\hat{x})\|] \le \epsilon$ with the iteration complexity of $T = O(d^3/\epsilon^2)$, where the expectation is taken w.r.t. random vectors $\{u_k\}$ and $\{v_k\}$. Further, if we choose $\gamma \le d^{-\Omega(\epsilon^2/d)}$, the iteration complexity can be bounded as $T = O(d/\epsilon^2)$.*

**Theorem C.2** (**Convergence of ZOSLGH with error tolerance, Stochastic setting**). *Suppose Assumptions A1, A2 and A3 hold. Take $k_1 := \Theta(d/\epsilon^4)$ and $k_2 := O\left(\log_\gamma 1/d\right)$ and define $k_0 = \min\{k_1, k_2\}$. Let $\hat{x} := x_{k'}$, where $k'$ is chosen from a uniform distribution over $\{k_0 + 1, k_0 + 2, \ldots, T\}$. Set the stepsize for $x$ at iteration $k$ as $\beta_k = \min\left\{\frac{1}{16(d+4)L_1(t_k)}, \frac{1}{\sqrt{(T-k_0)(d+4)}}\right\}$.*

*Then, for any setting of the parameter $\gamma$, if the error level $\nu$ satisfies $\nu = O(\epsilon^2/d^3)$, $\hat{x}$ satisfies $\mathbb{E}[\|\nabla f(\hat{x})\|] \le \epsilon$ with the iteration complexity of $T = O(d^2/\epsilon^4 + d^3/\epsilon^2)$, where the expectation is taken w.r.t. random vectors $\{u_k\}, \{v_k\}$ and $\{\xi_k\}$. Further, if we choose $\gamma \le d^{-\Omega(\epsilon^4/d)}$, the iteration complexity can be bounded as $T = O(d/\epsilon^4)$.*

### C.1 Proofs for technical lemmas

We introduce several lemmas before going to the convergence analysis. All of them describe properties of the function with error $f'$ and its Gaussian smoothing $F'$. Throughout this subsection, we assume that $f$ is $L_0$-Lipschitz and $L_1$-smooth function. We also suppose that the function pair $(f, f')$ satisfies $\sup_{x \in \mathbb{R}^d} |f(x) - f'(x)| \le \nu$.

**Lemma C.3.** *For any $x \in \mathbb{R}^d$ and $t > 0$, we have*

$$\mathbb{E}_u\left[\frac{1}{t^2}(f'(x+tu) - f'(x))^2\|u\|^2\right] \le 4(d+4)\|\nabla f(x)\|^2 + t^2 L_1^2 (d+6)^3 + 8d\frac{\nu^2}{t^2}.$$

**Proof:**

$$\mathbb{E}_u\left[\frac{1}{t^2}(f'(x+tu) - f'(x))^2\|u\|^2\right] = \mathbb{E}_u\left[\frac{1}{t^2}(f(x+tu) - f(x) + (f'-f)(x+tu) - (f'-f)(x))^2\|u\|^2\right]$$

$$\le 2\mathbb{E}_u\left[\frac{1}{t^2}(f(x+tu) - f(x))^2\|u\|^2\right] + 2\mathbb{E}_u\left[\frac{1}{t^2}(2\nu)^2\|u\|^2\right]$$

$$\le 4(d+4)\|\nabla f(x)\|^2 + t^2 L_1^2(d+6)^3 + 8d\frac{\nu^2}{t^2},$$

where the last inequality holds due to Lemma B.1 and Lemma B.4.

**Lemma C.4.** *For any $x \in \mathbb{R}^d$ and $t > 0$, we have*

$$\mathbb{E}_\zeta\left[\frac{1}{t^2}(\bar{f}'(x+tu;\xi) - \bar{f}'(x;\xi))^2\|u\|^2\right] \le 4(d+4)(\|\nabla f(x)\|^2 + \sigma^2) + t^2 L_1^2 (d+6)^3 + 8d\frac{\nu^2}{t^2}.$$

**Proof:**

$$\mathbb{E}_\zeta \left[ \frac{1}{t^2} (\bar{f}'(x+tu;\xi) - \bar{f}'(x;\xi))^2 \|u\|^2 \right]$$

$$= \mathbb{E}_\xi \left[ \mathbb{E}_u \left[ \frac{1}{t^2} (\bar{f}(x+tu;\xi) - \bar{f}(x;\xi) + (\bar{f}' - \bar{f})(x+tu;\xi) - (\bar{f}' - \bar{f})(x;\xi))^2 \|u\|^2 \right] \right]$$

$$\leq 2\mathbb{E}_\xi \left[ \mathbb{E}_u \left[ \frac{1}{t^2} (\bar{f}(x+tu;\xi) - \bar{f}(x;\xi))^2 \|u\|^2 \right] \right] + \frac{2}{t^2} \mathbb{E}_\xi [\mathbb{E}_u [(2\nu)^2 \|u\|^2]]$$

$$\leq 2\mathbb{E}_\xi \left[ \frac{t^2}{2} L_1^2 (d+6)^3 + 2(d+4)\|\nabla \bar{f}(x;\xi)\|^2 \right] + 8d \frac{\nu^2}{t^2}$$

$$\leq 4(d+4)(\|\nabla f(x)\|^2 + \sigma^2) + t^2 L_1^2 (d+6)^3 + 8d \frac{\nu^2}{t^2},$$

where the second inequality follows from Lemma B.1 and Lemma B.4, and the last inequality holds due to Assumption A2 (ii).

**Lemma C.5.** *For any $x \in \mathbb{R}^d$ and for any $t_1, t_2 \in \mathcal{T}$, we have*

$$|F'(x,t_1) - F'(x,t_2)| \leq L_0 |t_1 - t_2| \sqrt{d} + 2\nu.$$

**Proof:**

$$
\begin{aligned}
|F'(x,t_1) - F'(x,t_2)| &= |F(x,t_1) - F(x,t_2) + (F' - F)(x,t_1) - (F' - F)(x,t_2)| \\
&\leq |F(x,t_1) - F(x,t_2)| + |\mathbb{E}_u[(f' - f)(x+t_1 u)]| + |\mathbb{E}_u[(f' - f)(x+t_2 u)]| \\
&\leq |F(x,t_1) - F(x,t_2)| + \mathbb{E}_u[|(f' - f)(x+t_1 u)|] + \mathbb{E}_u[|(f' - f)(x+t_2 u)|] \\
&\leq |F(x,t_1) - F(x,t_2)| + 2\nu \\
&\leq L_0 |t_1 - t_2| \sqrt{d} + 2\nu,
\end{aligned}
$$

where the last inequality holds due to Lemma 3.3.

**Lemma C.6** (**Lemma 30 in [17]**). *For any $x \in \mathbb{R}^d$ and for any $t_1, t_2 \in \mathcal{T}$, we have*

$$\|\nabla_x (F' - F)(x,t)\| \leq \sqrt{\frac{2}{\pi}} \frac{\nu}{t}.$$

**Lemma C.7.**
(i) $F'(x,t)$ is $L_0 + \sqrt{\frac{2}{\pi}} \frac{\nu}{t}$-*Lipschitz in terms of $x$.*
(ii) (**Lemma 20 in [17]**) $F'(x,t)$ is $L_1 + \frac{2\nu}{t^2}$-*smooth in terms of $x$.*

**Proof for (i):**

$$\|\nabla_x F'(x,t)\| \leq \|\nabla_x F(x,t)\| + \|\nabla_x (F' - F)(x,t)\|$$

$$\leq L_0 + \sqrt{\frac{2}{\pi}} \frac{\nu}{t},$$

where the last inequality holds due to Lemma 3.2 and Lemma C.6.

**Lemma C.8.** *For any $x \in \mathbb{R}^d$ and $t > 0$, we have*

$$\|\nabla f(x)\|^2 \leq 4\|\nabla_x F'(x,t)\|^2 + \frac{t^2}{2} L_1^2 (d+6)^3 + \frac{8}{\pi} \frac{\nu^2}{t^2}.$$

**Proof:** We have

$$\|\nabla f(x)\|^2 = \|\mathbb{E}_u[\langle \nabla f(x), u\rangle u]\|^2$$

$$= \left\| \frac{1}{t}\mathbb{E}_u[(f(x+tu) - f(x) - [f(x+tu) - f(x) - t\langle \nabla f(x), u\rangle])u] \right\|^2$$

$$\leq \left\| \nabla_x F(x,t) - \frac{1}{t}\mathbb{E}_u[(f(x+tu) - f(x) - t\langle \nabla f(x), u\rangle)u] \right\|^2$$

$$\leq 2\|\nabla_x F(x,t)\|^2 + \frac{2}{t^2}\|\mathbb{E}_u[(f(x+tu) - f(x) - t\langle \nabla f(x), u\rangle)u]\|^2$$

$$\leq 2\|\nabla_x F(x,t)\|^2 + \frac{2}{t^2}\mathbb{E}_u[|f(x+tu) - f(x) - t\langle \nabla f(x), u\rangle|^2\|u\|^2]$$

$$\leq 2\|\nabla_x F(x,t)\|^2 + \frac{t^2 L_1^2}{2}\mathbb{E}_u[\|u\|^6]$$

$$\leq 2\|\nabla_x F(x,t)\|^2 + \frac{t^2 L_1^2}{2}(d+6)^3$$

$$\leq 2(2\|\nabla_x(F - F')(x,t)\|^2 + 2\|\nabla_x F'(x,t)\|^2) + \frac{t^2 L_1^2}{2}(d+6)^3$$

$$\leq 4\|\nabla_x F'(x,t)\|^2 + \frac{t^2 L_1^2}{2}(d+6)^3 + \frac{8}{\pi}\frac{\nu^2}{t^2},$$

where the third last inequality holds due to Lemma B.1, and the last inequality holds due to Lemma C.6.

## C.2 Proof for the deterministic setting

**Proof for Theorem C.1:** Let $w_k := (u_k, v_k)$ and denote $\delta_k := \tilde{g}'_x(x_k, t_k; u_k) - \nabla_x F'(x_k, t_k)$. Utilize the updating rule for $x$ and $L_1(t)$-smoothness of $F'(\cdot, t)$. Then we have

$$F'(x_{k+1}, t_k) \leq F'(x_k, t_k) + \langle \nabla_x F'(x_k, t_k), (x_{k+1} - x_k)\rangle + \frac{L_1(t_k)}{2}\|x_{k+1} - x_k\|^2$$

$$= F'(x_k, t_k) - \beta_k \langle \nabla_x F'(x_k, t_k), \tilde{g}'_x(x_k, t_k; u_k)\rangle + \frac{L_1(t_k)}{2}\beta_k^2\|\tilde{g}'_x(x_k, t_k; u_k)\|^2$$

$$= F'(x_k, t_k) - \beta_k\|\nabla_x F'(x_k, t_k)\|^2 - \beta_k \langle \nabla_x F'(x_k, t_k), \delta_k\rangle + \frac{L_1(t_k)}{2}\beta_k^2\|\tilde{g}'_x(x_k, t_k; u_k))\|^2.$$
(26)

Denote

$$E_k := -\beta_k \langle \nabla_x F'(x_k, t_k), \delta_k\rangle + \frac{L_1(t_k)}{2}\beta_k^2\|\tilde{g}'_x(x_k, t_k; u_k)\|^2$$

for simplicity. From Lemma C.5 and (26), we get the upper bound for $\|\nabla_x F'(x,t)\|^2$ as

$$\beta_k\|\nabla_x F'(x_k, t_k)\|^2 \leq F'(x_k, t_k) - F'(x_{k+1}, t_k) + E_k$$

$$= F'(x_k, t_k) - F'(x_{k+1}, t_{k+1}) + F'(x_{k+1}, t_{k+1}) - F'(x_{k+1}, t_k) + E_k$$

$$\leq F'(x_k, t_k) - F'(x_{k+1}, t_{k+1}) + L_0|t_{k+1} - t_k|\sqrt{d} + 2\nu + E_k.$$

Now, sum up the above inequality for all iterations $k_0 + 1 \le k \le T$ $(T > k_0)$. Then we have

$$\sum_{k=k_0+1}^{T} \beta_k \|\nabla_x F'(x_k, t_k)\|^2$$

$$\le F'(x_{k_0+1}, t_{k_0+1}) - F'(x_{T+1}, t_{T+1}) + L_0 \sum_{k=k_0+1}^{T} |t_{k+1} - t_k|\sqrt{d} + 2\nu(T - k_0) + \sum_{k=k_0+1}^{T} E_k$$

$$\le F'(x_{k_0+1}, t_{k_0+1}) - f^* + \nu + L_0\sqrt{d} \sum_{k=k_0+1}^{T} |t_{k+1} - t_k| + 2\nu(T - k_0) + \sum_{k=k_0+1}^{T} E_k.$$

$$\le f'(x_{k_0+1}) - f^* + 3\nu + L_0\sqrt{d} \left( t_{k_0+1} + \sum_{k=k_0+1}^{T} |t_{k+1} - t_k| \right) + 2\nu(T - k_0) + \sum_{k=k_0+1}^{T} E_k$$

$$\le f(x_{k_0+1}) - f^* + 4\nu + L_0\sqrt{d} \left( t_{k_0+1} + \sum_{k=k_0+1}^{T} |t_{k+1} - t_k| \right) + 2\nu(T - k_0) + \sum_{k=k_0+1}^{T} E_k,$$

$$\tag{27}$$

where the third inequality holds due to Lemma C.5. We can bound the conditional expectation of $E_k$ as

$$\mathbb{E}_{w_k}[E_k \mid w_{[k-1]}]$$

$$= -\beta_k \mathbb{E}_{w_k}[\langle \nabla_x F'(x_k, t_k), \delta_k \rangle \mid w_{[k-1]}] + \frac{\mathbb{E}_{w_k}[L_1(t_k) \mid w_{[k-1]}]}{2} \beta_k^2 \mathbb{E}_{w_k}[\|\tilde{g}_x'(x_k, t_k; u_k)\|^2 \mid w_{[k-1]}]$$

$$\le \frac{\mathbb{E}_{w_k}[L_1(t_k) \mid w_{[k-1]}]}{2} \beta_k^2 \mathbb{E}_{w_k}[\|\tilde{g}_x'(x_k, t_k; u_k)\|^2 \mid w_{[k-1]}]$$

$$\le \frac{\mathbb{E}_{w_k}[L_1(t_k) \mid w_{[k-1]}]}{2} \beta_k^2 \left( 4(d+4)\mathbb{E}_{w_k}[\|\nabla f(x_k)\|^2 \mid w_{[k-1]}] + L_1^2(d+6)^3 \mathbb{E}_{w_k}[t_k^2 \mid w_{[k-1]}] \right.$$

$$\left. + 8d\mathbb{E}_{w_k}[\nu^2/t_k^2 \mid w_{[k-1]}] \right),$$

where the first inequality holds since we have $\mathbb{E}_{w_k}[\delta_k \mid w_{[k-1]}] = \mathbb{E}_{u_k}[\delta_k \mid u_{[k-1]}] = 0$, and the last inequality holds due to Lemma C.3. Take the expectations of (27) w.r.t. random vectors $\{w_{k_0+1}, ..., w_T\}$. Then we can get

$$\sum_{k=k_0+1}^{T} \beta_k \mathbb{E}_w[\|\nabla_x F'(x_k, t_k)\|^2]$$

$$\le f(x_{k_0+1}) - f^* + 4\nu + L_0\sqrt{d} \left( t_{k_0+1} + \sum_{k=k_0+1}^{T} \mathbb{E}_w[|t_{k+1} - t_k|] \right) + 2\nu(T - k_0)$$

$$+ \frac{1}{2} \left( 4(d+4) \sum_{k=k_0+1}^{T} \beta_k^2 \mathbb{E}_w[L_1(t_k)\|\nabla f(x_k)\|^2] + L_1^2(d+6)^3 \sum_{k=k_0+1}^{T} \beta_k^2 \mathbb{E}_w[L_1(t_k)t_k^2] \right.$$

$$\left. + 8d \sum_{k=k_0+1}^{T} \beta_k^2 \mathbb{E}_w\left[ L_1(t_k)\frac{\nu^2}{t_k^2} \right] \right).$$

$$\tag{28}$$

Lemma C.8 together with (28) yields

$$\sum_{k=k_0+1}^{T} \beta_k \mathbb{E}_w[\|\nabla f(x_k)\|^2]$$

$$\leq 4 \sum_{k=k_0+1}^{T} \beta_k \mathbb{E}_w[\|\nabla_x F'(x_k, t_k)\|^2] + \frac{1}{2} \sum_{k=k_0+1}^{T} \beta_k \mathbb{E}_w[t_k^2] L_1^2 (d+6)^3 + \frac{8}{\pi} \sum_{k=k_0+1}^{T} \beta_k \mathbb{E}_w\left[\frac{\nu^2}{t_k^2}\right]$$

$$\leq 4 \left( f(x_{k_0+1}) - f^* + 4\nu + L_0\sqrt{d} \left( t_{k_0+1} + \sum_{k=k_0+1}^{T} \mathbb{E}_w[|t_{k+1} - t_k|] \right) + 2\nu(T - k_0) \right)$$

$$+ 2 \left( 4(d+4) \sum_{k=k_0+1}^{T} \beta_k^2 \mathbb{E}_w[L_1(t_k)\|\nabla f(x_k)\|^2] + L_1^2(d+6)^3 \sum_{k=k_0+1}^{T} \beta_k^2 \mathbb{E}_w[L_1(t_k)t_k^2] \right.$$

$$\left. +8d \sum_{k=k_0+1}^{T} \beta_k^2 \mathbb{E}_w\left[L_1(t_k)\frac{\nu^2}{t_k^2}\right] \right)$$

$$+ \frac{1}{2} \sum_{k=k_0+1}^{T} \beta_k \mathbb{E}_w[t_k^2] L_1^2(d+6)^3 + \frac{8}{\pi} \sum_{k=k_0+1}^{T} \beta_k \mathbb{E}_w\left[\frac{\nu^2}{t_k^2}\right].$$

By rearranging the terms, we obtain

$$\sum_{k=k_0+1}^{T} \left( \beta_k \mathbb{E}_w[\|\nabla f(x_k)\|^2] - 8(d+4)\beta_k^2 \mathbb{E}_w[L_1(t_k)\|\nabla f(x_k)\|^2] \right)$$

$$\leq 4 \left( f(x_{k_0+1}) - f^* + 4\nu + L_0\sqrt{d} \left( t_{k_0+1} + \sum_{k=k_0+1}^{T} \mathbb{E}_w[|t_{k+1} - t_k|] \right) + 2\nu(T - k_0) \right)$$

$$+ 2 \left( L_1^2(d+6)^3 \sum_{k=k_0+1}^{T} \beta_k^2 \mathbb{E}_w[L_1(t_k)t_k^2] + 8d \sum_{k=k_0+1}^{T} \beta_k^2 \mathbb{E}_w\left[L_1(t_k)\frac{\nu^2}{t_k^2}\right] \right)$$

$$+ \frac{1}{2} \sum_{k=k_0+1}^{T} \beta_k \mathbb{E}_w[t_k^2] L_1^2(d+6)^3 + \frac{8}{\pi} \sum_{k=k_0+1}^{T} \beta_k \mathbb{E}_w\left[\frac{\nu^2}{t_k^2}\right]. \tag{29}$$

If we update $t_k$ ($k \in [T]$) as in Algorithm 4, we have $\nu = O(t_k^2)$, which yields $L_1(t_k) = O(1)$ from Lemma C.7. Hence, by setting the step size $\beta_k$ as $\frac{1}{16(d+4)L_1(t_k)}$ ($k \in [T]$), we can obtain

$$\frac{1}{T - k_0} \sum_{k=k_0+1}^{T} \mathbb{E}_w[\|\nabla f(x_k)\|^2] = O\left( \frac{d}{T - k_0} \left( 1 + \sqrt{d} \sum_{k=k_0+1}^{T} \mathbb{E}_w[|t_{k+1} - t_k|] + d^2 \sum_{k=k_0+1}^{T} \mathbb{E}_w[t_k^2] \right) \right)$$

in the same way as before. We can also get $\sum_{k=k_0+1}^{T} |t_{k+1} - t_k| = \sum_{k=k_0+1}^{T}(t_k - t_{k+1}) = t_{k_0+1} - t_{T+1} = t_{k_0+1} = O(\gamma^{k_0})$. Further, we have

$$\sum_{k=k_0+1}^{T} t_k^2 \leq \sum_{k=k_0+1}^{T} \max\{t_1^2 \gamma^{2(k-1)}, \nu\} \leq \sum_{k=k_0+1}^{T} \left( t_1^2 \gamma^{2(k-1)} + \nu \right) \leq \frac{t_1^2 \gamma^{2k_0}}{1 - \gamma^2} + \nu(T - k_0)$$

$$= O(\gamma^{2k_0} + \nu(T - k_0)),$$

where the first inequality follows from the update rule of $t_k$ in Algorithm 4. Hence, we obtain

$$\frac{1}{T - k_0} \sum_{k=k_0+1}^{T} \mathbb{E}_w[\|\nabla f(x_k)\|^2] = O\left( \frac{d}{T - k_0} \left( 1 + \sqrt{d}\gamma^{k_0} + d^2(\gamma^{2k_0} + \nu(T - k_0)) \right) \right)$$

$$= O\left( \frac{d(1 + d^2\gamma^{2k_0})}{T - k_0} + d^3\nu \right) = O\left( \frac{d(1 + d^2\gamma^{2k_0})}{T - k_0} + \epsilon^2 \right),$$

where the last equality follows from the assumption of $\nu = O(\epsilon^2/d^3)$.

Here, we have $k_0 = O\left(\frac{d}{\epsilon^2}\right)$ by the definition of $k_0$. Thus, by setting $T = k_0 + O\left(\frac{d^3}{\epsilon^2}\right) = O\left(\frac{d^3}{\epsilon^2}\right)$, we can obtain $\mathbb{E}_{w,k'}[\|\nabla f(\hat{x})\|^2] = \frac{1}{T-k_0}\sum_{k=k_0+1}^T \mathbb{E}_w[\|\nabla f(x_k)\|^2] \leq \epsilon^2$. This implies $\mathbb{E}_{w,k'}[\|\nabla f(\hat{x})\|] \leq \epsilon$ as $\mathbb{E}_{w,k'}[\|\nabla f(\hat{x})\|]^2 \leq \mathbb{E}_{w,k'}[\|\nabla f(\hat{x})\|^2]$ follows from Jensen's inequality. Furthermore, when $\gamma$ is chosen as $\gamma \leq d^{-\epsilon^2/d}$, we have $\log_\gamma d^{-1} = O\left(\frac{d}{\epsilon^2}\right)$, which implies $k_0 = \Omega\left(\log_\gamma d^{-1}\right)$. Therefore, we can obtain $\gamma^{k_0} = O(d^{-1})$, which yields the iteration complexity of $T = O\left(\frac{d}{\epsilon^2}\right)$.

$\square$

## C.3  Proof for the stochastic setting

**Proof for Theorem C.2:**

Let $\zeta_k := (\xi_k, u_k, v_k)$, $k \in [T]$ and denote $\delta_k := \tilde{G}'_x(x_k, t_k; \xi_k, u_k) - \nabla_x F'(x_k, t_k)$. Since $\tilde{G}'_x(x, t; \xi, u)$ is an unbiased estimator of $\nabla_x F'(x, t)$, we have

$$F'(x_{k+1}, t_k) \leq F'(x_k, t_k) + \langle \nabla_x F'(x_k, t_k), (x_{k+1} - x_k)\rangle + \frac{L_1(t_k)}{2}\|x_{k+1} - x_k\|^2$$

$$= F'(x_k, t_k) - \beta_k \left\langle \nabla_x F'(x_k, t_k), \tilde{G}'_x(x_k, t_k; \xi_k, u_k)\right\rangle + \frac{L_1(t_k)}{2}\beta_k^2\|\tilde{G}'_x(x_k, t_k; \xi_k, u_k)\|^2$$

$$= F'(x_k, t_k) - \beta_k\|\nabla_x F'(x_k, t_k)\|^2 - \beta_k \langle \nabla_x F'(x_k, t_k), \delta_k\rangle + \frac{L_1(t_k)}{2}\beta_k^2\|\tilde{G}'_x(x_k, t_k; \xi_k, u_k)\|^2.$$

Now, denote

$$I_k := -\beta_k \langle \nabla_x F'(x_k, t_k), \delta_k\rangle + \frac{L_1(t_k)}{2}\beta_k^2\|\tilde{G}'_x(x_k, t_k; \xi_k, u_k)\|^2$$

for simplicity. Then, we can get the upper bound for $\|\nabla_x F(x,t)\|^2$ with $I_k$:

$$\beta_k\|\nabla_x F'(x_k, t_k)\|^2 \leq F'(x_k, t_k) - F'(x_{k+1}, t_k) + I_k$$

$$= F'(x_k, t_k) - F'(x_{k+1}, t_{k+1}) + F'(x_{k+1}, t_{k+1}) - F'(x_{k+1}, t_k) + I_k$$

$$\leq F'(x_k, t_k) - F'(x_{k+1}, t_{k+1}) + L_0|t_{k+1} - t_k|\sqrt{d} + 2\nu + I_k,$$

where the last inequality follows from Lemma C.5. Sum up the above inequality for all iterations $k_0 + 1 \leq k \leq T$. Then we have

$$\sum_{k=k_0+1}^T \beta_k\|\nabla_x F'(x_k, t_k)\|^2$$

$$\leq F'(x_{k_0+1}, t_{k_0+1}) - F'(x_{T+1}, t_{T+1}) + L_0\sqrt{d}\sum_{k=k_0+1}^T |t_{k+1} - t_k| + 2\nu(T - k_0) + \sum_{k=k_0+1}^T I_k$$

$$\leq f(x_{k_0+1}) - f^* + 4\nu + L_0\sqrt{d}\left(t_{k_0+1} + \sum_{k=k_0+1}^T |t_{k+1} - t_k|\right) + 2\nu(T - k_0) + \sum_{k=k_0+1}^T I_k.$$
(30)

We can also obtain

$$\mathbb{E}_{\zeta_k}[I_k \mid \zeta_{[k-1]}]$$

$$= -\beta_k\mathbb{E}_{\zeta_k}[\langle \nabla_x F'(x_k, t_k), \delta_k\rangle \mid \zeta_{[k-1]}] + \frac{\mathbb{E}_{\zeta_k}[L_1(t_k) \mid \zeta_{[k-1]}]}{2}\beta_k^2\mathbb{E}_{\zeta_k}[\|\tilde{G}'_x(x_k, t_k; \xi_k, u_k)\|^2 \mid \zeta_{[k-1]}]$$

$$= \frac{\mathbb{E}_{\zeta_k}[L_1(t_k) \mid \zeta_{[k-1]}]}{2}\beta_k^2\mathbb{E}_{\zeta_k}[\|\tilde{G}'_x(x_k, t_k; \xi_k, u_k)\|^2 \mid \zeta_{[k-1]}]$$

$$\leq \frac{\mathbb{E}_{\zeta_k}[L_1(t_k) \mid \zeta_{[k-1]}]s}{2}\beta_k^2\left(4(d + 4)(\mathbb{E}_{\zeta_k}[\|\nabla f(x_k)\|^2 \mid \zeta_{[k-1]}] + \sigma^2) + \mathbb{E}_{\zeta_k}[t_k^2 \mid \zeta_{[k-1]}]L_1^2(d + 6)^3\right.$$

$$\left. + 8d\mathbb{E}_{\zeta_k}\left[\nu^2/t_k^2 \mid \zeta_{[k-1]}\right]\right),$$

where the last inequality holds due to Lemma C.4.

Take the expectation of (30) with respect to $\zeta_{k_0+1}, \ldots, \zeta_T$. Then we have

$$\sum_{k=k_0+1}^{T} \beta_k \mathbb{E}_\zeta[\|\nabla_x F'(x_k, t_k)\|^2]$$

$$\leq f(x_{k_0+1}) - f^* + 4\nu + L_0\sqrt{d}\left(t_{k_0+1} + \sum_{k=k_0+1}^{T} \mathbb{E}_\zeta[|t_{k+1} - t_k|]\right) + 2\nu(T - k_0) + \sum_{k=k_0+1}^{T} \mathbb{E}_\zeta[I_k]$$

$$\leq f(x_{k_0+1}) - f^* + 4\nu + L_0\sqrt{d}\left(t_{k_0+1} + \sum_{k=k_0+1}^{T} \mathbb{E}_\zeta[|t_{k+1} - t_k|]\right) + 2\nu(T - k_0)+$$

$$+ \frac{1}{2}\left(4(d+4)\sum_{k=k_0+1}^{T} \beta_k^2(\mathbb{E}_\zeta[L_1(t_k)\|\nabla f(x_k)\|^2] + \sigma^2) + L_1^2(d+6)^3\sum_{k=k_0+1}^{T} \beta_k^2\mathbb{E}_\zeta[L_1(t_k)t_k^2]\right.$$

$$\left.+ 8d\sum_{k=k_0+1}^{T} \beta_k^2\mathbb{E}_\zeta\left[L_1(t_k)\frac{\nu^2}{t_k^2}\right]\right),$$

From Lemma C.8 (ii), we have

$$\sum_{k=k_0+1}^{T} \beta_k \mathbb{E}_\zeta[\|\nabla f(x_k)\|^2]$$

$$\leq 4\sum_{k=k_0+1}^{T} \beta_k \mathbb{E}_\zeta[\|\nabla_x F'(x_k, t_k)\|^2] + \frac{L_1^2(d+6)^3}{2}\sum_{k=k_0+1}^{T} \beta_k \mathbb{E}_\zeta[t_k^2] + \frac{8}{\pi}\sum_{k=k_0+1}^{T} \beta_k \mathbb{E}_\zeta\left[\frac{\nu^2}{t_k^2}\right]$$

$$\leq 4\left(f(x_{k_0+1}) - f^* + 4\nu + L_0\sqrt{d}\left(t_{k_0+1} + \sum_{k=k_0+1}^{T} \mathbb{E}_\zeta[|t_{k+1} - t_k|]\right) + 2\nu(T - k_0)\right)$$

$$+ 2\left(4(d+4)\sum_{k=k_0+1}^{T} (\beta_k^2\mathbb{E}_\zeta[L_1(t_k)(\|\nabla f(x_k)\|^2 + \sigma^2)]) + L_1^2(d+6)^3\sum_{k=k_0+1}^{T} \mathbb{E}_\zeta\beta_k^2[L_1(t_k)t_k^2]\right.$$

$$\left.+ 8d\sum_{k=k_0+1}^{T} \beta_k^2\mathbb{E}_\zeta\left[L_1(t_k)\frac{\nu^2}{t_k^2}\right]\right)$$

$$+ \frac{L_1^2(d+6)^3}{2}\sum_{k=k_0+1}^{T} \beta_k \mathbb{E}_\zeta[t_k^2] + \frac{8}{\pi}\sum_{k=k_0+1}^{T} \beta_k \mathbb{E}_\zeta\left[\frac{\nu^2}{t_k^2}\right]. \tag{31}$$

By rearranging the terms, we obtain

$$\sum_{k=k_0+1}^{T} \left(\beta_k \mathbb{E}_\zeta[\|\nabla f(x_k)\|^2] - 8(d+4)\beta_k^2\mathbb{E}_\zeta[L_1(t_k)\|\nabla f(x_k)\|^2]\right)$$

$$\leq 4\left(f(x_{k_0+1}) - f^* + 4\nu + L_0\sqrt{d}\left(t_{k_0+1} + \sum_{k=k_0+1}^{T} \mathbb{E}_\zeta[|t_{k+1} - t_k|]\right) + 2\nu(T - k_0)\right)$$

$$+ 2\left(4(d+4)\sigma^2\sum_{k=k_0+1}^{T} \beta_k^2\mathbb{E}_\zeta[L_1(t_k)] + L_1^2(d+6)^3\sum_{k=k_0+1}^{T} \beta_k\mathbb{E}_\zeta[L_1(t_k)t_k^2] + 8d\sum_{k=k_0+1}^{T} \beta_k\mathbb{E}_\zeta\left[L_1(t_k)\frac{\nu^2}{t_k^2}\right]\right)$$

$$+ \frac{L_1^2(d+6)^3}{2}\sum_{k=k_0+1}^{T} \beta_k \mathbb{E}_\zeta[t_k^2] + \frac{8}{\pi}\sum_{k=k_0+1}^{T} \beta_k \mathbb{E}_\zeta\left[\frac{\nu^2}{t_k^2}\right]. \tag{32}$$

If we update $t_k$ $(k \in [T])$ as in Algorithm 4, we have $\nu = O(t_k^2)$, which yields $L_1(t_k) = O(1)$ from Lemma C.7. Furthermore, if we set the step size $\beta_k$ as $\min\left\{\frac{1}{16(d+4)L_1(t_k)}, \frac{1}{\sqrt{(T-k_0)(d+4)}}\right\}$ $(k \in$

$[T])$, then we have

$$\frac{1}{\beta_k - 8(d+4)L_1(t_k)\beta_k^2} \leq \frac{2}{\beta_k},$$

$$\frac{1}{\beta_k} \leq 16(d+4)L_1(t_k) + \sqrt{(T-k_0)(d+4)}.$$

for all $k \in [T]$. Using the above inequalities, we can obtain

$$\frac{1}{T-k_0}\sum_{k=k_0+1}^{T}\mathbb{E}_\zeta[\|\nabla f(x_k)\|^2] = O\left(\frac{\sqrt{d}\left(1+\sqrt{d}\sum_{k=k_0+1}^{T}\mathbb{E}_\zeta[|t_{k+1}-t_k|]\right)}{\sqrt{T-k_0}} + \frac{d\left(1+d^2\sum_{k=k_0+1}^{T}\mathbb{E}_\zeta[t_k^2]\right)}{T-k_0}\right)$$

$$= O\left(\frac{\sqrt{d}+d\gamma^{k_0}}{\sqrt{T-k_0}} + \frac{d+d^3\gamma^{2k_0}}{T-k_0} + d^3\nu\right)$$

$$= O\left(\frac{\sqrt{d}+d\gamma^{k_0}}{\sqrt{T-k_0}} + \frac{d+d^3\gamma^{2k_0}}{T-k_0} + \epsilon^2\right),$$

where the second and last equality can be shown via a similar way as in the proof of Theorem C.1.

Here, we have $k_0 = O\left(\frac{d}{\epsilon^4}\right)$ by the definition of $k_0$. Thus, by setting $T = O\left(\frac{d^3}{\epsilon^2} + \frac{d^2}{\epsilon^4}\right) = O\left(\frac{d^3}{\epsilon^2} + \frac{d^2}{\epsilon^4}\right)$, we can obtain $\mathbb{E}_{\zeta,k'}[\|\nabla f(\hat{x})\|^2] = \frac{1}{T-k_0}\sum_{k=k_0+1}^{T}\mathbb{E}_\zeta[\|\nabla f(x_k)\|^2] \leq \epsilon^2$. This implies $\mathbb{E}_{\zeta,k'}[\|\nabla f(\hat{x})\|] \leq \epsilon$ as $\mathbb{E}_{\zeta,k'}[\|\nabla f(\hat{x})\|]^2 \leq \mathbb{E}_{\zeta,k'}[\|\nabla f(\hat{x})\|^2]$ follows from Jensen's inequality. Furthermore, when $\gamma$ is chosen as $\gamma \leq d^{-\epsilon^4/d}$, we have $\log_\gamma d^{-1} = O\left(\frac{d}{\epsilon^4}\right)$, which implies that $k_0 = \Omega\left(\log_\gamma d^{-1}\right)$. Therefore, we can obtain $\gamma^{k_0} = O(d^{-1})$, which yields the iteration complexity of $T = O\left(\frac{d}{\epsilon^4}\right)$.

$\square$

# D    Optimization of test functions

In the first three subsections, let us compare the performance of our SLGH algorithms with GD-based algorithms and double loop GH algorithms using highly-non-convex test functions for optimization: the Ackley function [26], Rosenbrock function, and Himmelblau function [1]. We implemented the following five types of algorithms: (ZOS)GD, (ZO)GradOpt, in which the factor for decreasing the smoothing parameter was 0.5 or 0.8, (ZO)SLGH$_r$ with $\gamma = 0.995$ or $\gamma = 0.999$.

## D.1    Ackley Function

The Ackley function is defined as

$$f(x,y) = -20\exp\left[-0.2\sqrt{0.5\left(x^2+y^2\right)}\right] - \exp[0.5(\cos 2\pi x + \cos 2\pi y)] + e + 20,$$

whose global optimum is $f(0,0) = 0$. As shown in Figure 3(a), it has numerous small local minima due to cosine functions which are included in the second term. We ran the aforementioned five types of zeroth-order algorithms with the stepsize $\beta = 0.1$ for $T = 1000$ iterations. The initial smoothing parameter for the GH algorithms (ZOGradOpt and ZOSLGH$_r$) was set to $t_1 = 1$, where local minima of the smoothed function almost disappeared (Figure 3(b)). The smoothing parameter for ZOSGD was chosen as $t = 0.005$. We set the initial point for the optimization as $(x,y) = (5,5)$.

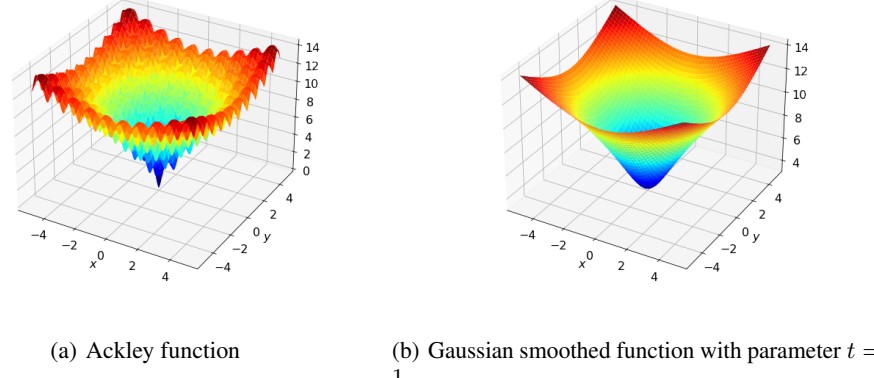

(a) Ackley function

(b) Gaussian smoothed function with parameter $t = 1$

Figure 3: Visualization of the Ackley function and its Gaussian smoothed function.

We illustrate the optimization results in Table 3 and Figure 4. The GH methods successfully reach near the optimal solution $(0, 0)$ when the decreasing speed of $t$ is not so fast, while ZOSGD is stuck in a local minimum in the immediate vicinity of the initial point $(5, 5)$. Please note that GradOpt succeeds in optimization without decreasing the smoothing parameter since the optimal solution of the smoothed function with $t = 1$ almost matches that of the original target function.

Table 3: Optimization results of the Ackley function. The global optimum is $f(0, 0) = 0$.

|  | Methods | $(x, y)$ | $f(x, y)$ |
|---|---|---|---|
| SGD algo. | ZOSGD | $(4.99, 4.99)$ | 12.63 |
| GH algo. | ZOGradOpt $(\gamma = 0.5)$ | $(4.2 \times 10^{-3}, 1.9 \times 10^{-3})$ | $\mathbf{1.4 \times 10^{-2}}$ |
|  | ZOGradOpt $(\gamma = 0.8)$ | $(-2.2 \times 10^{-3}, 6.7 \times 10^{-3})$ | $\mathbf{8.1 \times 10^{-2}}$ |
|  | ZOSLGH$_\mathrm{r}$ $(\gamma = 0.995)$ | $(1.97, 1.97)$ | 6.56 |
|  | ZOSLGH$_\mathrm{r}$ $(\gamma = 0.999)$ | $(-3.6 \times 10^{-3}, -4.6 \times 10^{-3})$ | $\mathbf{1.7 \times 10^{-2}}$ |

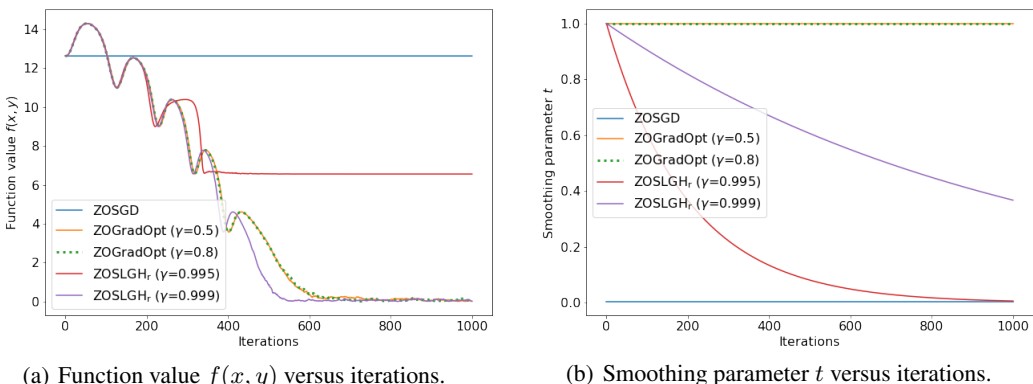

(a) Function value $f(x, y)$ versus iterations.

(b) Smoothing parameter $t$ versus iterations.

Figure 4: Plots of the function value and the smoothing parameter during optimization of the Ackley function.

## D.2 Rosenbrock Function

Let us define the Rosenbrock function in 2D as

$$f(x, y) = 100 \left(y - x^2\right)^2 + (1 - x)^2,$$

whose global optimum is $f(1, 1) = 0$. This function is difficult to optimize because the global optimum lies inside a flat parabolic shaped valley with low function value (Figure 5(a)). Since this function is polynomial, we can calculate the GH smoothed function analytically (see [25]):

$$F(x, y, t) := \mathbb{E}_{u_x, u_y}[f(x + tu_x, y + tu_y)], \quad (u_x, u_y \sim \mathcal{N}(0, 1))$$
$$= 100x^4 + (-200y + 600t^2 + 1)x^2 - 2x + 100y^2 - 200t^2y + (300t^4 + 101t^2 + 1).$$

Thus, we applied first-order methods to this function. The stepsize and iteration number were set to $\beta = 1 \times 10^{-4}$ and $T = 20000$, respectively. The initial smoothing parameter for the GH algorithms (GradOpt and SLGH$_r$) was set to $t_1 = 1.5$, where the smoothed function became almost convex around the optimal solution (Figure 5(b)). We set the initial point for the optimization as $(x, y) = (-3, 2)$.

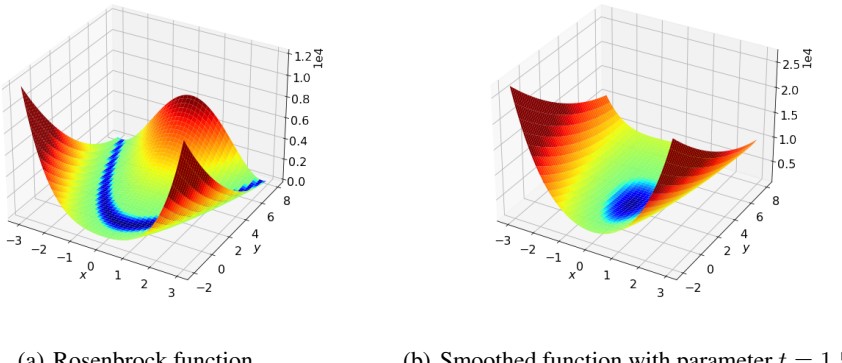

(a) Rosenbrock function        (b) Smoothed function with parameter $t = 1.5$

Figure 5: Visualization of the Rosenbrock function and its Gaussian smoothed function.

We illustrate the optimization results in Table 4, Figure 6 and Figure 7. The GH methods can decrease the function value much faster than GD. This is because the smoothed function is much easier to optimize than the original function while its optimal solution is close to that of the original one. In the early stage of optimization, the GH methods reach near a point $(0, 2)$, which is a good initial point for optimization, while GD falls into a point in the flat valley, which is far from the optimal solution. (Figure 7).

Table 4: Optimization results of the Rosenbrock function. The global optimum is $f(1, 1) = 0$.

| | Methods | $(x, y)$ | $f(x, y)$ |
|---|---|---|---|
| GD algo. | GD | $(0.468, 0.216)$ | $0.284$ |
| GH algo. | GradOpt ($\gamma = 0.5$) | $(0.817, 0.667)$ | $\mathbf{3.36 \times 10^{-2}}$ |
| | GradOpt ($\gamma = 0.8$) | $(0.808, 0.652)$ | $\mathbf{3.70 \times 10^{-2}}$ |
| | SLGH$_r$ ($\gamma = 0.995$) | $(0.819, 0.670)$ | $\mathbf{3.27 \times 10^{-2}}$ |
| | SLGH$_r$ ($\gamma = 0.999$) | $(0.795, 0.631)$ | $\mathbf{4.19 \times 10^{-2}}$ |

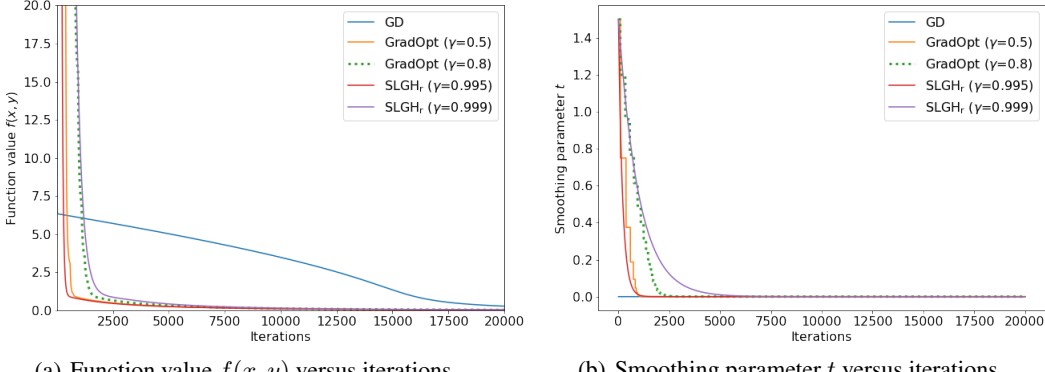

(a) Function value $f(x, y)$ versus iterations.    (b) Smoothing parameter $t$ versus iterations.

Figure 6: Plots of the function value and the smoothing parameter during optimization of the Rosenbrock function.

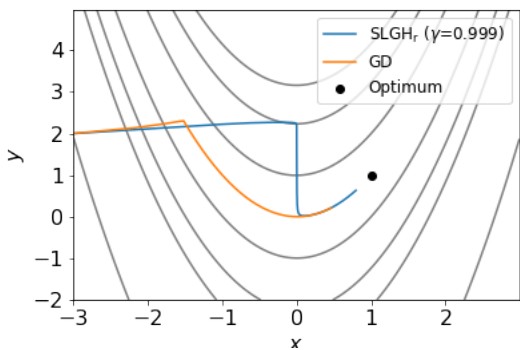

Figure 7:    Comparison of output sequences between GD and SLGH$_r$ ($\gamma = 0.999$) with contours of the Rosenbrock function.

### D.3   Himmelblau Function

The Himmelblau function is defined as

$$f(x, y) = (x^2 + y - 11)^2 + (x + y^2 - 7)^2.$$

It has four minimum points in the vicinity of $(x, y) = (3.000, 2.000), (-2.805, 3.131), (-3.779, -3.283), (3.584, -1.848)$ and one maximum point in the vicinity of $(x, y) = (-0.271, -0.923)$. It takes the optimal value 0 at the four points. Since this function is also polynomial, we can calculate the GH smoothed function analytically:

$$F(x, y, t) := \mathbb{E}_{u_x, u_y}[f(x + tu_x, y + tu_y)], \quad (u_x, u_y \sim \mathcal{N}(0, 1))$$
$$= x^4 + (2y + 6t^2 - 21)x^2 + (2y^2 + 2t^2 - 14)x + y^4 + (6t^2 - 13)y^2 + (2t^2 - 22)y + (6t^4 - 34t^2 + 170).$$

Thus, we applied first-order methods to this function. The stepsize and iteration number were set to $\beta = 1 \times 10^{-4}$ and $T = 2000$, respectively. The initial smoothing parameter for GH algorithms was set to $t_1 = 2$, where the smoothed function became almost convex around the optimal solution (Figure 8(b)). We set the initial point for the optimization as $(x, y) = (5, 5)$.

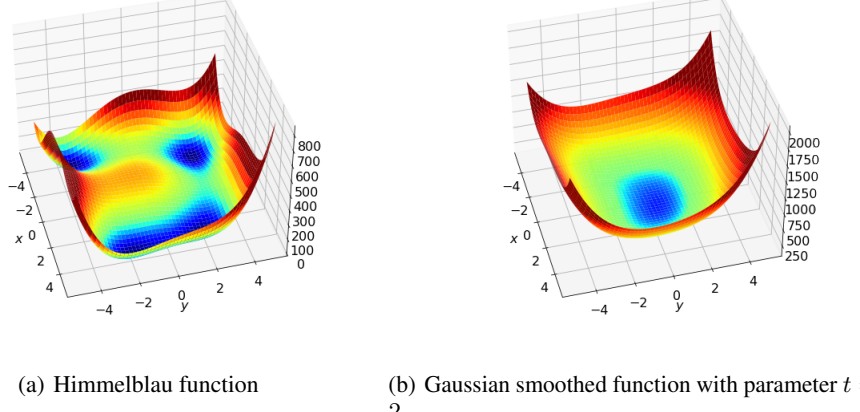

(a) Himmelblau function

(b) Gaussian smoothed function with parameter $t = 2$

Figure 8: Visualization of the Himmelblau function and its Gaussian smoothed function.

Table 5, Figure 9, and Figure 10 show the optimization results. GD and our SLGH algorithms successfully reach near the global optimum, while GradOpt fails to decrease the function value. This is because the optimal solution of the smoothed function when $t = 2$ lies near the maximum point of the original Himmelblau function $(-0.271, -0.923)$. Figure 10 describes detailed optimization process. Our SLGH algorithm succeeds in returning to the optimal solution once it has passed by reducing $t$. In contrast, GradOpt reaches the vicinity of a minimum of the smoothed function without knowing the detailed shape of the original function; as a result, it is stuck around a local maximum of the original function.

Table 5: Results of optimization of the Himmelblau function. It has a global optimum $f(3, 2) = 0$.

|  | Methods | $(x, y)$ | $f(x, y)$ |
|---|---|---|---|
| GD algo. | GD | $(2.998, 2.003)$ | $\mathbf{1.6 \times 10^{-4}}$ |
| GH algo. | GradOpt ($\gamma = 0.5$) | $(2.575, 1.437)$ | $14.14$ |
|  | GradOpt ($\gamma = 0.8$) | $(1.573, 0.868)$ | $80.51$ |
|  | SLGH$_\mathrm{r}$ ($\gamma = 0.995$) | $(2.999, 2.002)$ | $\mathbf{6.9 \times 10^{-5}}$ |
|  | SLGH$_\mathrm{r}$ ($\gamma = 0.999$) | $(2.983, 1.897)$ | $\mathbf{0.21}$ |

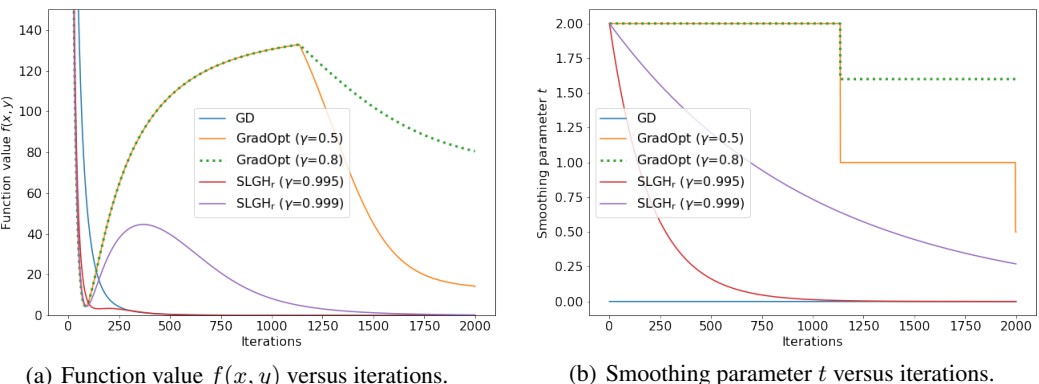

(a) Function value $f(x, y)$ versus iterations.

(b) Smoothing parameter $t$ versus iterations.

Figure 9: Plots of the function value and the smoothing parameter during optimization of the Himmelblau function.

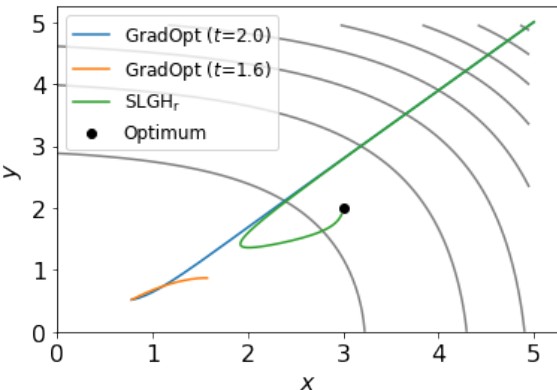

Figure 10: Comparison of output sequences of GradOpt, in which the factor for decreasing the smoothing parameter is 0.8, and SLGH$_r$ ($\gamma = 0.999$) with contours of the *smoothed* Himmelblau function. The blue and orange lines represent output sequences of GradOpt when $t = 2.0$ and $t = 1.6$, respectively.

### D.4   Additional Toy Example

At the end of this section, let us present a toy example problem in which SLGH$_d$, which utilizes the derivative $\frac{\partial F}{\partial t}$ for the update of $t$, outperforms SLGH$_d$. Let us consider the following artificial non-convex function:

$$ f(x, y) = \begin{cases} x^2 - 150 \times 1.1^{-((x-10)^2 + y^2)} & (x \geq 0) \\ x^2/50 - 150 \times 1.1^{-((x-10)^2 + y^2)} & (x < 0) \end{cases} . $$

The second term creates a hole around $(x, y) = (10, 0)$ (see Figure 12(a)), and this function has an optimum in the vicinity of $f(9.319, 0) \simeq -56.670$. This function is difficult to optimize for GH methods since the hole around the optimum disappears when the smoothing parameter $t$ is large (Figure 12(b)).

We ran SLGH$_r$ ($\gamma = 0.995$ or $0.999$) and SLGH$_d$ ($\gamma = 0.999$) with the stepsize (for $x$) $\beta = 0.01$ for $T = 1000$ iterations. The initial point and initial smoothing parameter were set to $(x, y) = (15, 0)$ and $t_1 = 5$, respectively. We set the stepsize for $t$ as $0.01$.

Table 6 and Figure 11 show the optimization results. We can see that only SLGH$_d$ can decrease $t$ around the hole adaptively, and thus successfully can find the optimal solution.

Table 6: Optimization results of the artificial non-convex function. It has a global optimum in the vicinity of $f(9.319, 0) \simeq -56.670$.

| | Methods | $(x, y)$ | $f(x, y)$ |
|---|---|---|---|
| GH algo. | SLGH$_r$ ($\gamma = 0.995$) | $(-0.248, 2.38 \times 10^{-2})$ | $-5.52 \times 10^{-3}$ |
| | SLGH$_r$ ($\gamma = 0.999$) | $(-2.959, -2.18 \times 10^{-3})$ | $0.175$ |
| | SLGH$_d$ ($\gamma = 0.999$) | $(9.319, 8.33 \times 10^{-3})$ | $\mathbf{-56.670}$ |

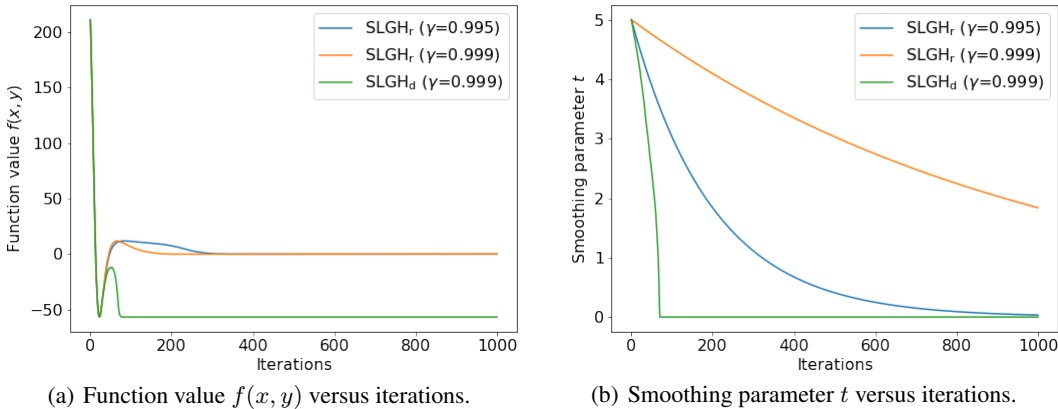

(a) Function value $f(x, y)$ versus iterations.

(b) Smoothing parameter $t$ versus iterations.

Figure 11: Plots of the function value and the smoothing parameter during optimization of the artificial non-convex function.

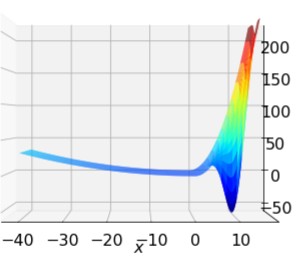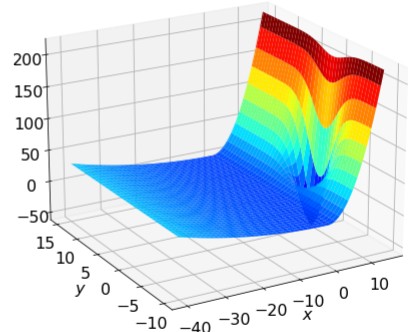

(a) Artificial non-convex function

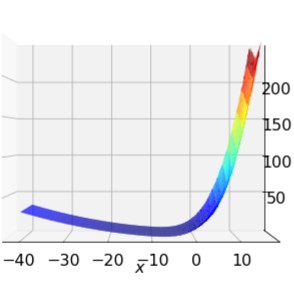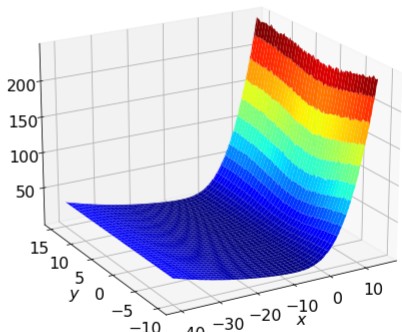

(b) Gaussian smoothed function with parameter $t = 5$

Figure 12: Visualization of the artificial non-convex function and its Gaussian smoothed function.

# E  Black-box adversarial attack

## E.1  Experimental Setup

We used well-trained DNNs[4] for CIFAR10 and MNIST classification tasks as target models, respectively. We adopt the implementation[5] in [9] for ZOSGD and ZOAdaMM. GradOpt [14] in our implementation adopts the same random gradient-free oracles [28] as with our ZOSLGH methods, rather than their smoothed gradient oracle, where random variables are sampled from the unit sphere. Moreover, we set the stepsize in its inner loop as a constant instead of $\Theta(1/k)$, where $k$ denotes an iteration number in the inner loop, due to less efficiency of the original setting. Therefore, the essential difference between GradOpt and ZOSLGH$_r$ is whether or not the structure of algorithms is single loop.

As recommended in their work, we set the parameter for ZOAdaMM as $v_0 = 10^{-5}$, $\beta_1 = 0.9$, and $\beta_2 = 0.3$. The factor for decreasing the smoothing parameter in ZOGradOpt was set to 0.5. For all algorithms, we chose the regularization parameter $\lambda$ as $\lambda = 10$ and set attack confidence $\kappa = 1e - 10$. We chose minibatch size as $M = 10$ to stabilize estimation of values and gradients of the smoothed function. The initial adversarial perturbation was chosen as $x_0 = 0$, and the initial smoothing parameter $t_0$ was 10 for GH methods and 0.005 for the others. The decreasing factor for $t$ in the ZOSLGH algorithm was set to $\gamma = 0.999$ for both of ZOSLGH$_r$ and ZOSLGH$_d$, unless otherwise noted. Other parameter settings are described in Table 7. We used different step sizes for ZOAdaMM because it adaptively penalizes the step size using the information of past gradients [9].

Table 7: Parameter settings in the adversarial attack problems. $T$ represents the iteration number. $\beta$ is the step size for $x$, and $\eta$ is the step size for $t$. $N_0$ and $\epsilon_0$ are used to determine termination condition of the inner loop in ZOGradOpt: we stop the inner loop and decrease $t$ if the condition $|\frac{1}{M}\sum_{i=1}^{M} f(x_{k+1} + tu_i) - \frac{1}{M}\sum_{i=1}^{M} f(x_k + tu_i')| \leq \epsilon_0$ is satisfied $N_0$ times, where $u_i$ and $u_i'$ ($i = 1, ..., M$) are sampled from $\mathcal{N}(0, \mathrm{I}_d)$. Each of "3072" and "784" is the dimension of images in CIFAR-10 and MNIST.

| | $T$ | $\beta$ (other than ZOAdaMM) | $\beta$ (for ZOAdaMM) | $\eta$ | $(N_0, \epsilon_0)$ |
|---|---|---|---|---|---|
| CIFAR-10 | 10000 | 0.01/3072 | 0.5/3072 | $1 \times 10^{-4}/3072$ | $(100, 5 \times 10^{-3})$ |
| MNIST | 20000 | 1/784 | 100/784 | 0.1/784 | $(100, 1 \times 10^{-3})$ |

## E.2  CIFAR-10

**Additional plots**  Figures 13 and 14 show additional plots for total loss and $L_2$ distortion, respectively. We can see that our ZOSLGH algorithms successfully decrease the total loss value except in cases where images are so difficult to attack that no algorithms succeed in attacking (Figure 13(i), 13(j)). Plots in Figure 14 imply that the algorithms are stuck around a local minimum $x = 0$ when they are failed to decrease the loss value.

---

[4]https://github.com/carlini/nn_robust_attacks
[5]https://github.com/KaidiXu/ZO-AdaMM

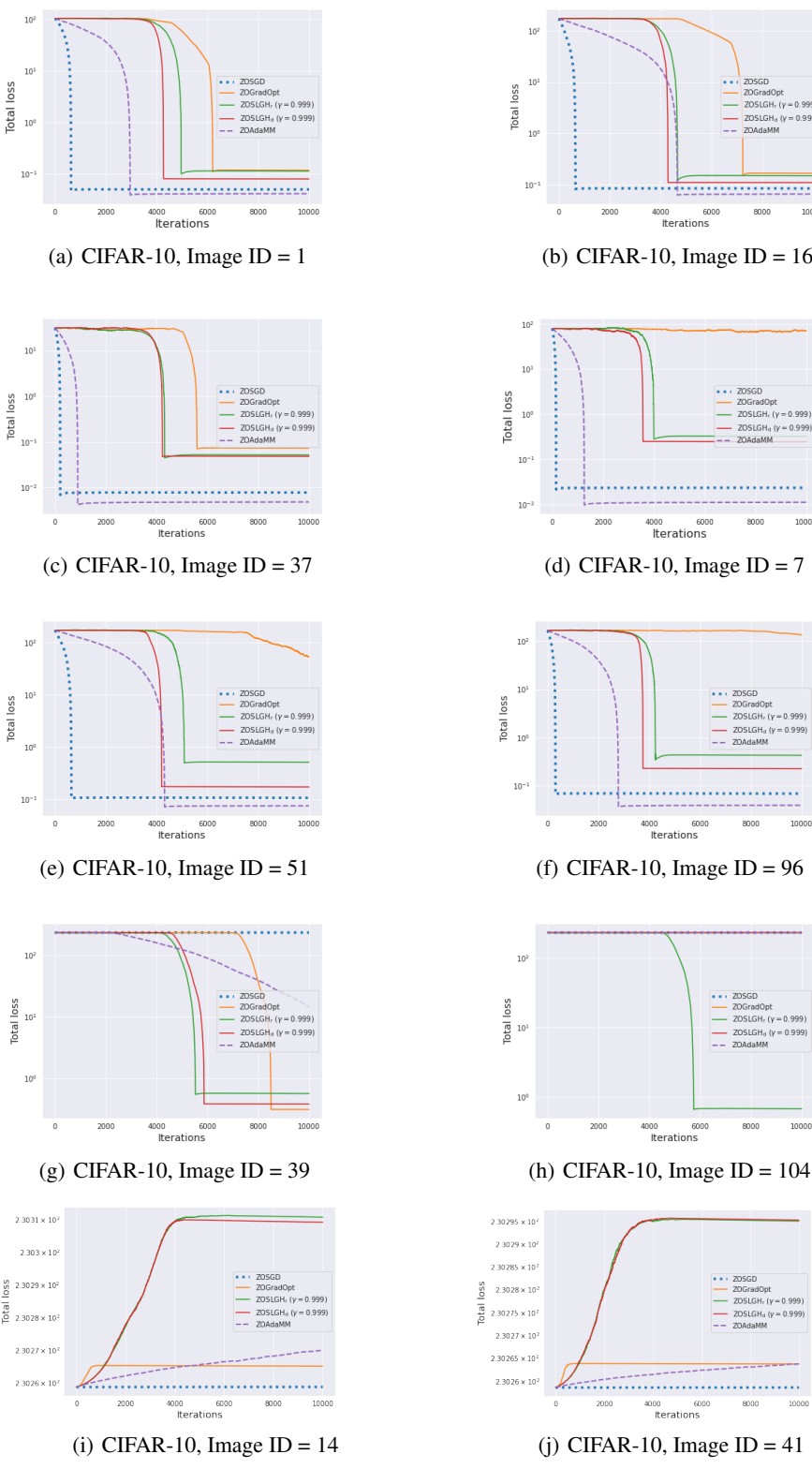

Figure 13: Additional plots of total loss versus iterations on CIFAR-10 (log scale). (a)-(c) All algorithms can successfully decrease the loss value when images are easy to attack. In particular, in plot (c), SGD-based algorithms can find better solutions than GH-based algorithms. (d)-(f) Only GradOpt fails to attack due to its slow convergence. (g) Only ZOSGD is stuck around a local minimum $x = 0$. (h) Only our ZOSLGH$_r$ algorithm succeeds in escaping the local minimum, and thus it can decrease the loss value more than 200 than other algorithms. (i), (j): These images are so difficult to attack that no algorithms can succeed in attacking.

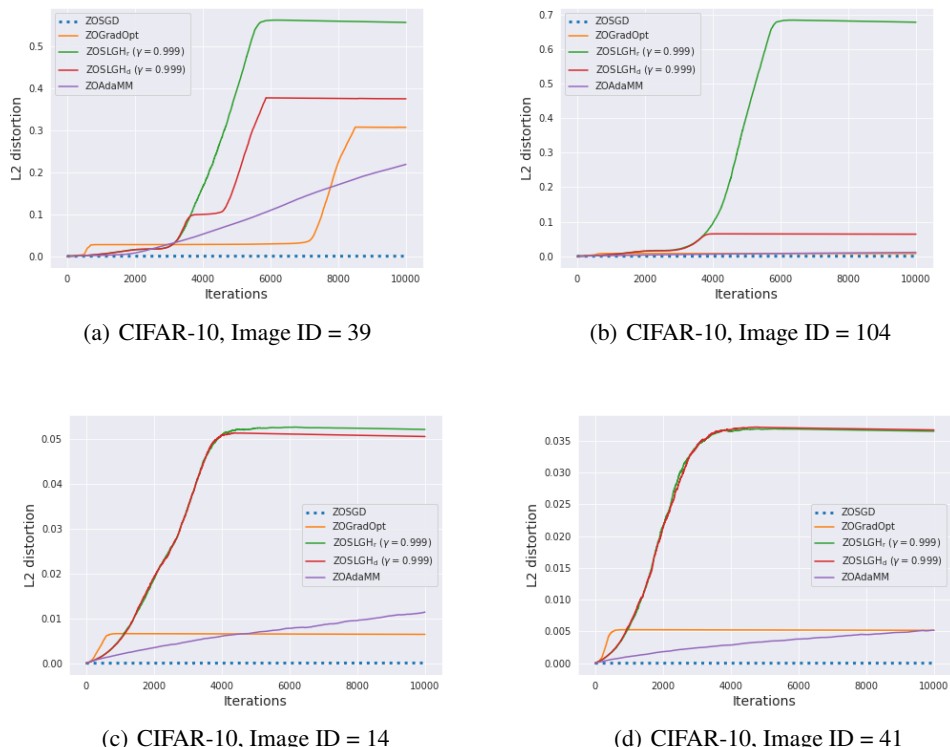

(a) CIFAR-10, Image ID = 39

(b) CIFAR-10, Image ID = 104

(c) CIFAR-10, Image ID = 14

(d) CIFAR-10, Image ID = 41

Figure 14: Plots of $L_2$ distortion versus iterations for images that are difficult to attack on CIFAR-10. Each plot of (a)-(d) corresponds to Figure 13(g)-Figure 13(j).

**Effect of choice of the parameter $\gamma$ in the ZOSLGH algorithm**   We also investigated the effect of choice of the decreasing parameter $\gamma$ in the ZOSLGH algorithm. We compared ZOSGD, ZOSLGH$_\text{r}$ with $\gamma = 0.995$, and ZOSLGH$_\text{r}$ with $\gamma = 0.999$. All other parameters were set to the same values as before. Figure 15 implies that the decreasing speed of $t$ is associated with a trade-off: a rapid decrease of $t$ yields fast convergence, but reduces the possibility to find better solutions.

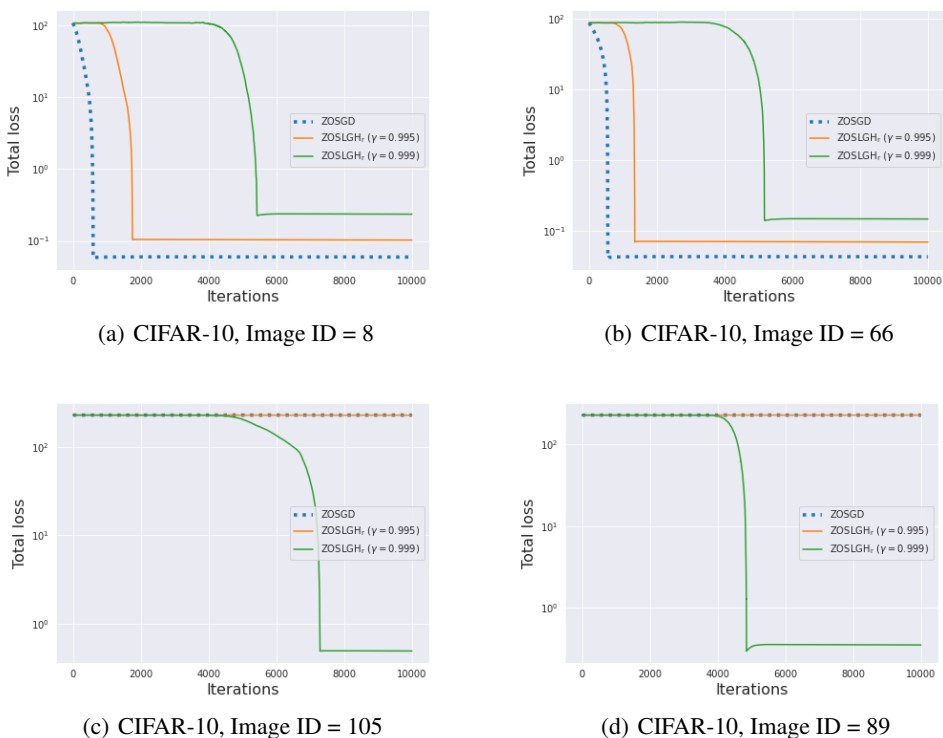

(a) CIFAR-10, Image ID = 8

(b) CIFAR-10, Image ID = 66

(c) CIFAR-10, Image ID = 105

(d) CIFAR-10, Image ID = 89

Figure 15: Comparison of total loss transition of ZOSGD, ZOSLGH$_\text{r}$ with $\gamma = 0.995$, and ZOSLGH$_\text{r}$ with $\gamma = 0.999$ (log scale).

**Generated adversarial examples**  Table 8 shows adversarial images generated by different algorithms and their original images.

Table 8: Comparison of adversarial images for CIFAR-10 with different algorithms.

| Image ID | 39 | 79 | 89 | 115 |
|---|---|---|---|---|
| Original | | | | |
| Classified as | dog | ship | truck | cat |
| $L_2$ distortion: | 0 | 0 | 0 | 0 |
| ZOSGD | | | | |
| Classified as | dog (fail.) | airplane | truck (fail.) | horse |
| $L_2$ distortion: | $6.7 \times 10^{-5}$ | 0.154 | $5.6 \times 10^{-5}$ | $4.5 \times 10^{-3}$ |
| ZOAdaMM | | | | |
| Classified as | dog (fail.) | airplane | truck (fail.) | horse |
| $L_2$ distortion: | 0.226 | 0.145 | 0.131 | $1.6 \times 10^{-3}$ |
| ZOGradOpt | | | | |
| Classified as | cat | airplane | truck (fail.) | horse |
| $L_2$ distortion: | 0.304 | 0.254 | $1.1 \times 10^{-30}$ | 0.192 |
| ZOSLGH$_r$ | | | | |
| Classified as | cat | airplane | automobile | horse |
| $L_2$ distortion: | 0.540 | 0.212 | 0.282 | 0.076 |
| ZOSLGH$_d$ | | | | |
| Classified as | cat | airplane | automobile | horse |
| $L_2$ distortion: | 0.359 | 0.174 | 0.241 | 0.075 |

### E.3  MNIST

Finally, let us show the experimental results on the MNIST dataset. Our ZOSLGH algorithms attain higher success rates than other algorithms on this dataset as well as CIFAR-10 (Table 9). Moreover, the average number of iterations to achieve the first successful attack becomes comparable to ZOSGD. The main difference from the results on CIFAR-10 is that the average of $L_2$ distortion at successful time becomes far larger, from $0.050 \sim 0.250$ to $4.25 \sim 5.20$. This implies that attacks on MNIST are more difficult than those on CIFAR-10. See Figure 16 and Figure 17 for additional plots for total loss and $L_2$ distortion. Figure 10 shows adversarial images generated by different algorithms and their original images.

Table 9: Performance of a per-image attack over 100 images of MNIST under $T = 20000$ iterations. "Succ. rate" indicates the ratio of success attack, "Avg. iters to 1st succ." is the average number of iterations to reach the first successful attack, "Avg. $L_2$ (succ.)" is the average of $L_2$ distortion taken among successful attacks, and "Avg. total loss" is the average of total loss $f(x)$ over 100 samples. Please note that the standard deviations are large since the attack difficulty varies considerably from sample to sample.

|  | Methods | Succ. rate | Avg. iters to 1st succ. | Avg. $L_2$ (succ.) | Avg. total loss |
|---|---|---|---|---|---|
| SGD algo. | ZOSGD | 67% | $1171 \pm 1954$ | $4.83 \pm 4.13$ | $73.60 \pm 102.70$ |
|  | ZOAdaMM | 71% | $\mathbf{261} \pm 1068$ | $\mathbf{4.25} \pm 3.36$ | $67.49 \pm 100.25$ |
| GH algo. | ZOGradOpt | 84% | $6166 \pm 4354$ | $5.16 \pm 2.28$ | $28.25 \pm 65.35$ |
|  | ZOSLGH$_r$ ($\gamma = 0.999$) | $\mathbf{96\%}$ | $1537 \pm 277$ | $\mathbf{4.32} \pm 2.44$ | $\mathbf{11.83} \pm 37.88$ |
|  | ZOSLGH$_d$ ($\gamma = 0.999$) | $\mathbf{96\%}$ | $1342 \pm 242$ | $\mathbf{4.37} \pm 2.58$ | $\mathbf{12.09} \pm 38.56$ |

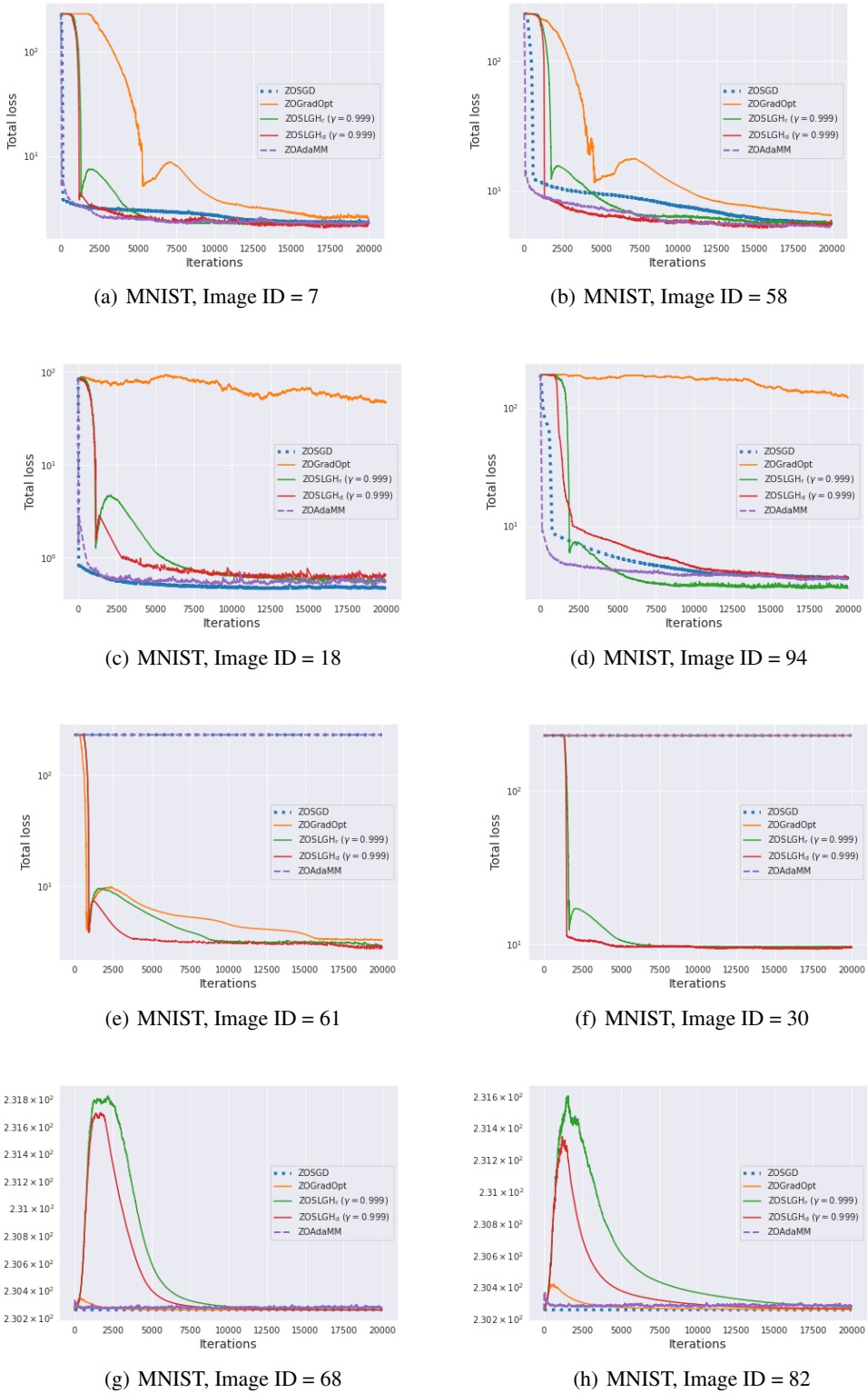

(a) MNIST, Image ID = 7

(b) MNIST, Image ID = 58

(c) MNIST, Image ID = 18

(d) MNIST, Image ID = 94

(e) MNIST, Image ID = 61

(f) MNIST, Image ID = 30

(g) MNIST, Image ID = 68

(h) MNIST, Image ID = 82

Figure 16: Additional plots of total loss versus iterations on MNIST (log scale). (a)-(b) All algorithms can successfully decrease the loss value when images are easy to attack. (c)-(d) Only GradOpt fails to attack due to its slow convergence. (e) ZOSGD and ZOAdaMM are stuck around a local minimum $x = 0$. (f) Only our ZOSLGH algorithms succeed in escaping the local minimum, and thus they can decrease the loss value more than 200 than other algorithms. (g), (h): These images are so difficult to attack that no algorithms can succeed in attacking.

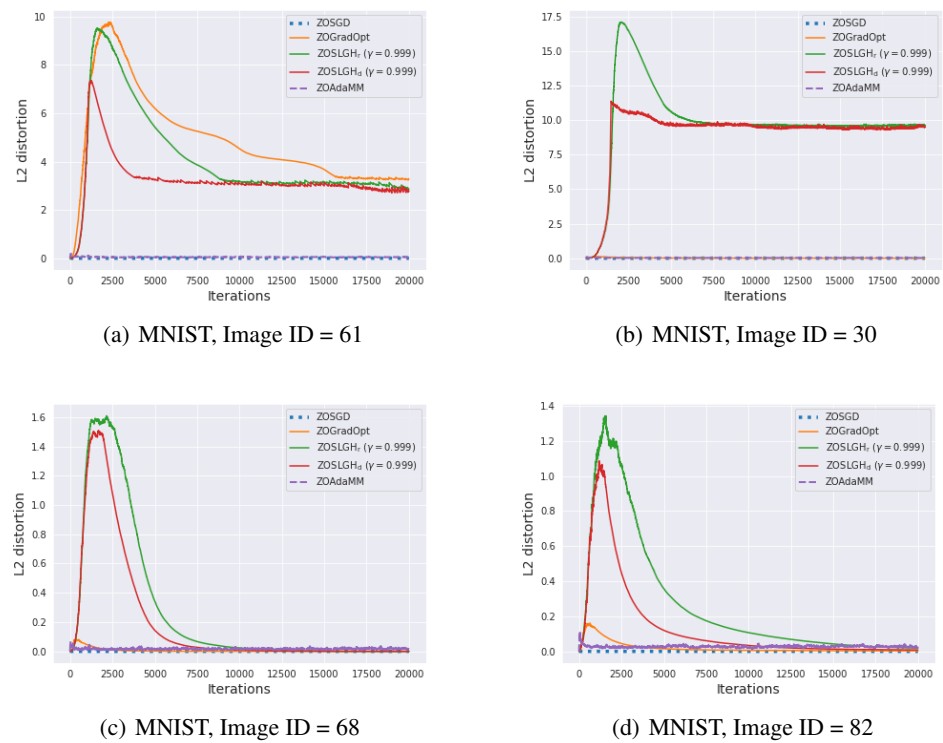

(a) MNIST, Image ID = 61

(b) MNIST, Image ID = 30

(c) MNIST, Image ID = 68

(d) MNIST, Image ID = 82

Figure 17: Plots of $L_2$ distortion versus iterations for images that are difficult to attack on MNIST. Each plot of (a)-(d) corresponds to Figure 16(e)-Figure 16(h).

Table 10: Comparison of the adversarial images for MNIST with different algorithms.

| Image ID | 10 | 21 | 48 | 83 |
|---|---|---|---|---|
| Original | | | | |
| Classified as | 0 | 6 | 4 | 7 |
| $L_2$ distortion: | 0 | 0 | 0 | 0 |
| ZOSGD | | | | |
| Classified as | 0 (fail.) | 5 | 9 | 7 (fail.) |
| $L_2$ distortion: | $4.1 \times 10^{-7}$ | 1.194 | 1.183 | $1.8 \times 10^{-4}$ |
| ZOAdaMM | | | | |
| Classified as | 0 (fail.) | 5 | 9 | 7 (fail.) |
| $L_2$ distortion: | $4.9 \times 10^{-14}$ | 1.334 | 1.100 | $4.0 \times 10^{-14}$ |
| ZOGradOpt | | | | |
| Classified as | 2 | 5 | 9 | 9 |
| $L_2$ distortion: | 3.898 | 1.378 | 1.903 | 6.379 |
| ZOSLGH$_r$ | | | | |
| Classified as | 2 | 5 | 9 | 9 |
| $L_2$ distortion: | 3.867 | 1.261 | 1.106 | 6.075 |
| ZOSLGH$_d$ | | | | |
| Classified as | 2 | 5 | 9 | 9 |
| $L_2$ distortion: | 4.048 | 1.222 | 1.059 | 5.722 |