# OpenReview forum: "Single Loop Gaussian Homotopy Method for Non-convex Optimization"
_NeurIPS.cc/2022/Conference — NeurIPS 2022 Accept_

### Official Review · Reviewer_JQgj · 2022-07-11

**Rating:** 7
**Confidence:** 4
**Soundness:** 3 good
**Presentation:** 3 good
**Contribution:** 3 good

**Summary:**

In this work, the authors studied the Gaussian homotopy (GH) method for non-convex optimization problems in deterministic and stochastic settings. Compared to the existing GH-based method with the two-loop scheme, they proposed a single-loop GD method that can update the smoothing parameter and optimization variable at the same time. In particular, they showed the computational complexity of the proposed methods in different settings. Finally, experimental results are reported to demonstrate the potential of the proposed methods.

**Questions:**


(${\bf 1}$) The idea of the Gaussian smoothed function in Definition 2.1 is very similar to that in Eq (17) of [8]. Could the authors elaborate on the connections and differences between the gaussian homotopy method and the smoothing gradient descent method in [8] and the related paper, like the iteration complexity?

(${\bf 2}$) In the introduction part, the authors mentioned that "The Gaussian homotopy is designed to avoid poor stationary points...". However, in the remaining part of the paper, it seems that the authors cannot illustrate this advantage of the GH method. Could the authors demonstrate this point theoretically or empirically?

(${\bf 3}$) The smoothing methods usually employ the fixed-ratio update rule for updating the smoothing parameter $t$. In Section 3.2, the authors proposed an interesting derivative update rule. Compared to the fixed-ratio update rule, what is the benefit of the derivative update rule? Can this significantly improve algorithm performance?

(${\bf 4}$) Could the proposed method be easily extended to the constrained setting?


**Strengths And Weaknesses:**


Strengths: (${\bf 1}$)  This work has a good contribution to the Gaussian homotopy (GH) methods. The authors not only developed a single-loop GD method but also extended it to the stochastic and zeroth-order settings. Moreover, the complexity analysis of the proposed methods is provided.

(${\bf 2}$) Overall, the paper is well-written and I think that the authors made good enough technical contributions.

Weaknesses: (${\bf 1}$) It seems that Assumption A.1 (i) is rather strict, which requires the objective function $f$ to be bounded. Many functions arising in applications don't satisfy this assumption. Thus, this really limits the applications of the GH methods. Note that in [8], no such assumption is imposed. Could this assumption be relaxed?

(${\bf 2}$) In Figure (1), to better compare function values of the tested method, the authors should use the log-scale plot on the y-axis. From the current figures, the loss function values of most of the tested methods look the same.

---

> ### Author Response · Authors · 2022-08-02
> **Thank you for supporting our paper and providing constructive comments. We address your detailed questions below.**
>
> Q1: It seems that Assumption A.1 (i) is rather strict, which requires the objective function $f$ to be bounded. Many functions arising in applications don't satisfy this assumption. Thus, this really limits the applications of the GH methods. Note that in [8], no such assumption is imposed. Could this assumption be relaxed?
>
> A1:
> - The boundedness assumed in Assumption A.1 (i) is used to guarantee that the order of derivative in terms of $x$, integral in terms of $u$, and integral in terms of $\xi$ are exchangeable.
> - It can be certainly relaxed: it is sufficient if the following properties are satisfied
>     - Objective function $f$ satisfies $\sup_{x\in\mathbb{R}^d}\mathbb{E}_u[|f(x+tu)|] < \infty$.
>     - (In the stochastic setting,  $f$ satisfies $\sup_{x\in\mathbb{R}^d,\xi}\mathbb{E}_u[|\bar{f}(x+tu;\xi)|] < \infty$.)
> - We updated Assumption A.1 (i) and the proof for Lemma B.3 in the revised paper. Please see line 132 in the main paper and lines 101-102 for more details.
> - Please note that this assumption is mandatory to construct a zeroth-order gradient estimator using Gaussian smoothing.
>
> Q2: In Figure (1), to better compare function values of the tested method, the authors should use the log-scale plot on the y-axis. From the current figures, the loss function values of most of the tested methods look the same.
>
> A2:
> - Thank you for your opinion. We will change the scale of the axes in the final version.
>
>
>
>
> Q3: The idea of the Gaussian smoothed function in Definition 2.1 is very similar to that in Eq (17) of [8]. Could the authors elaborate on the connections and differences between the gaussian homotopy method and the smoothing gradient descent method in [8] and the related paper, like the iteration complexity?
>
> A3:
> - First, please note that [8] is another direction of study compared to ours. It smoothes non-smooth objective functions to provide smoothness with it. On the other hand, our study smooths the observed function mainly to find better points.
> - The smoothing function defined in Definition 1 in [8] includes a wide range of functions, and the GH function, which is used in this study, is also included. However, as described in line 19 in the appendix, [8] provided a construction method of only smoothing functions that can be reformulated by using the plus function $(t)_+ := \max\\{0, t\\}.$ Therefore, we need to develop smoothing methods individually for general non-convex functions. Furthermore, [8] did not analyze iteration complexity; it only proved that any accumulation point generated by the smoothing gradient method is a Clarke stationary point.
>
> Q4: In the introduction part, the authors mentioned that "The Gaussian homotopy is designed to avoid poor stationary points...". However, in the remaining part of the paper, it seems that the authors cannot illustrate this advantage of the GH method. Could the authors demonstrate this point theoretically or empirically?
>
> A4:
> - We empirically demonstrated that the proposed method could find better points than (ZO) SGD-based methods in many cases through numerical experiments. For example, see lines 296-301 in the main paper or Sections D, E in the appendix. Providing theoretical demonstration is one of our future works.
>
> Q5: The smoothing methods usually employ the fixed-ratio update rule for updating the smoothing parameter $t$. In Section 3.2, the authors proposed an interesting derivative update rule. Compared to the fixed-ratio update rule, what is the benefit of the derivative update rule? Can this significantly improve algorithm performance?
>
> A5:
> - The results on the toy problem in subsection D.4 in the appendix (in line 381-) show that, empirically, $\text{SLGH}_{\text{d}}$ has the potential to obtain better solutions by using derivative information in terms of $t$.
> -  $SLGH_d$ significantly decreases $t$ when the derivative of the GH function in terms of $t$ takes a large negative value, that is, when the value of GH function significantly decreases by slightly changing $t$ in the negative direction. Since the region around the optimal solution in the toy problem had such a structure, only $\text{SLGH}_{\text{d}}$ would be able to decrease $t$ around the hole and find the optimal solution.
> - To specify function classes for which $\text{SLGH}_{\text{d}}$ is superior to other methods will become one of the future works.
>
>
> Q6: Could the proposed method be easily extended to the constrained setting?
>
> A6:
> - We might be able to extend to the constrained setting by extending the projected gradient method. We would like to consider it in future work.

---

### Official Review · Reviewer_T91c · 2022-07-12

**Rating:** 5
**Confidence:** 3
**Soundness:** 2 fair
**Presentation:** 3 good
**Contribution:** 2 fair

**Summary:**

The paper proposes a single loop framework to deal with Gaussian Homotopy (GH) method for non-convex (deterministic and stochastic) optimization $f(x)$. GH approach aims to solve successively smooth surrogates $F(x,t)$ of $f(x)$ for $t \geq 0$ such that at convergence the minimization of $F(x,0)$ corresponds to a stationnary point of the initial problem. The proposed method coined SLGH (single loop GH) relies on a gaussian smoothing convolution $F(x,t)$ of $f$ that is simultaneously optimized w.r.t to $x$ and $t$ using a (projected-like) gradient descent. SLGH differs from classical approaches that optimizes $F(x,t)$ for pre-specified values of $t$. Two variants of SLGH are provided based respectively on first-order and zeroth-order gradient descent. Convergence analyses in terms of iteration complexity are conducted in each setting. Experimental evaluations on adversarial image crafting illustrate the behaviour of the methods.

**Questions:**

* In practice the choice of the step-sizes $\beta$  and $\eta$ may not be obvious. Is a line-search strategy used to appropriately estimate those parameters?
* Algorithms 2 and 3 use the parameter $\epsilon$. Is it the same parameter that is applied in Theorems 3.4, 3.5, 4.1 and 4.2?
* The paper claims that empirically, settings of $\gamma$ close to 1, e.g., $\gamma= 0.999$ work well in practice. Is such a choice efficient computationally? Because it probably induces a slow decrease of $t$ and hence a high iteration complexity.
* Why the same update policy is not applied to $t$ for SLGH_r in Algorithms 2 and 3?
* Computation of the Gaussian convolution $F(x,t)$ of $f(x)$ can be computationally demanding, rendering per iteration costly in Algorihm2. Are there typical obbjective functions that are suited for firs-order SLGH?
* When dealing with stochastic setting of SLGH, does the convergence improve when using a batch size greater than 1?
* Can the paper provide empirical evaluations of the methods illustrating the derived convergence rates? Also it would be interesting to show convergence results in terms of computation time and compared to plain gradient descent method for instance.
* Can the authors report fooling rate for different levels of L2 distortion to allow a fair and easy comparison of the methods?

**Limitations:**

The paper  discusses the broader impact of the proposed approaches. On my view point the work does not raise any specific issues.


**Strengths And Weaknesses:**

* The paper is well written and  well  organized. The rationale behind the proposed SLGH is justified and the new aspects brought by  the approach are stated clearly.

* As for a strength, the paper proposes a principled way to design the smoothing function $F(x,t)$ using Gaussian convolution applied to $f(x)$. It also establishes that the minimum of $F(x,t)$ w.r.t $t$ is achieved at $t=0$ giving then the solution to the minimization of $f(x)$. However computing $F(x,t)$ will be a hard task in general leaving the first order optimization method non-applicable in practice. This is mitigated by the proposition of a zeroth-order approach.

* The optimization scheme in Algorithm 2 is rather standard for solving $\min_{x, t \geq 0} F(x,t)$. The interesting fact is that $F(x,t)$ inherits boundness and Lipschitzness properties of $f$ that helps implementing Algorithm 2. Key results of the papers are the theoretical iteration complexities to reach an $\epsilon$ convergence on the gradient norm. Similar results are derived for zeroth-order methods. Generally speaking the first-order methods exhibit reduced iteration complexity.

* GH methods aim to provide solution to the original problem as the minimizer of $F(x,t=0)$ at convergence. However Algorithms 2 and 3 ouput the sought solution as a minimizer of $F(x,t=\epsilon)$ for some non-null positive $\epsilon$. It would be interesting to analyze convergence in the iterates $x_k$.

* Reported empirical evaluations in Section 5 are questionable and to argue for the proposed methods. As for the average number of iterations to reach the first successful attack, SGD algorithms perform better than GH methods. Regarding pediction peformance the paper reports the average of total loss $f(x)$ which is meaningless in the context of adversarial attack. The most relevant metric might be the fooling rate as one is interested in the success of attack not in the value of the loss function. Also the performances are compared for different levels of L2 distortion which is highly questionable. As a fair comparison, the considered methods should be evaluated for the same L2 distortion error.

* The authors provide relevant feedback to most reviewers' concerns. Nevertheless the reported experiments on adversarial attack in Section 5 do not prove the effectiveness of the GH approaches. Indeed,  the proposed GH methods lead to a smaller objective value compared to competitors (SGD algos). However they lead to higher fooling rates of the network but at the price of a higher perturbation. Therefore the attacks by GH cannot be claimed better than the ones by SGD algos.

---

> ### Author Response · Authors · 2022-08-02
> **Thank you for providing comments including useful suggestions. We answer your questions and comments below.**
>
> Q1: Is a line-search strategy used to appropriately estimate those parameters?
>
> A1: In practice, backtracking line search using a rule about the stepsize adoption (e.g. Armijo rule) may achieve better convergence than when using the fixed stepsize. Proving that is left to future work, although additional assumptions would be required.
>
> Q2: Algorithms 2 and 3 use the parameter $\epsilon$. Is it the same parameter that is applied in Theorems 3.4, 3.5, 4.1 and 4.2?
>
> A2:
> - $\epsilon$ used in the update rule of $t$ and that used in the error guarantee in the theorems are completely different. We modified their notations in the revised version.
> - Please also see the answer A1 for Reviewer WqLq to understand the role of $\epsilon$ used in the update rule of $t$.
>
> Q3: The paper claims that empirically, settings of $\gamma$ close to 1, e.g., $\gamma=0.999$ work well in practice. Is such a choice efficient computationally?
>
> A3: In our opinion, settings of $\gamma$ close to 1 would be still computationally efficient. The decrease of $t$ might be faster than Reviewer T91c expected even in such a case. Please note it holds that$$0.999^{5000}\simeq0.00672,\ 0.995^{1000}\simeq0.00665.$$
>
> Q4: Why the same update policy is not applied to $t$ for SLGH_r in Algorithms 2 and 3?
>
> A4: $\epsilon$ in the update rule of $t$ for SLGH_r was unnecessary. We fixed it in the revised paper.
>
> Q5: Are there typical objective functions that are suited for first-order SLGH?
>
> A5:
> - [22] shows some important family of functions for which the closed form of the GH function $F(x,t)$ can be calculated.
> - We added “Gaussian RBFs and trigonometric functions” in addition to "polynomial functions" as an example where the GH function is easily computable in the Introduction. Please see lines 73-75 in the revised main paper.
>
> Q6: When dealing with stochastic setting of SLGH, does the convergence improve when using a batch size greater than 1?
>
> A6: Yes, the convergence improves with increasing batchsize. The larger a batchsize is, the closer to 0 the expectation of variance of the gradient estimator is. Thereby, the iteration complexity becomes closer to that in the deterministic case.
>
> Q7: Can the paper provide empirical evaluations of the methods illustrating the derived convergence rates? Also it would be interesting to show convergence results in terms of computation time and compared to plain gradient descent method for instance.
>
> A7:
> - We implemented additional experiments to validate the derived convergence rates for Rosebrock and Himmelblau functions. SLGH_r seemed to achieve linear convergence, which is much better than the derived convergence rate based on the worst-case analysis. Please see “additional_expr” folder in the supplementary material for more details.
> - The computational time of the zeroth-order gradient estimator is dominant in the adversarial attack problem. Thus, SLGH_d took approximately twice the computational time per iteration as the other algorithm. The other algorithms had the almost same computational time per iteration.
>
> Q8: Can the authors report fooling rate for different levels of L2 distortion to allow a fair and easy comparison of the methods?
>
> A8:
> - Certainly, the attack success rate for the same L2 distortion error is one of the most important evaluation metrics in the context of the adversarial attack.
> - However, it is difficult to compare the performance of the algorithms for the same L2 distortion under the current formulation (see line 282 in the main paper). The first term with the coefficient $\lambda$ represents how well we fool the model, and the second L2 distortion represents the noise level. We need to tune $\lambda$ for each algorithm to match the value of L2 distortion, but it will become elaborate work.
> - We can also add constraints on the L2 distortion to match both the L2 distortion level and the computation time for all the algorithms. However, such a constrained optimization problem is beyond the scope of our work, and we want to leave it as future work.
> - Our experiment would view the optimization problem differently from Reviewer T91c. It compared the optimization algorithms based on which algorithm could decrease the value of the fixed objective function most (within a predetermined time).
> - Furthermore, our experiment may also seem to provide meaningful insight in the context of the adversarial attack. SLGH algorithms could achieve a higher success rate than SGD-based methods by escaping local minimum $x=0$ using smoothing. The minimum arose due to L2 distortion. This result implies that SLGH algorithms will be able to find good adversarial noise with a high ratio, even if we set the regularization coefficient $\lambda$ to a smaller value than the optimal one, and L2 distortion becomes more dominant. Thus, SLGH algorithms might be able to maintain high success rates without precisely tuning $\lambda$.

---

> > ### Comment · Reviewer_T91c · 2022-08-08
> > **Clarifications of the authors**
> >
> > Thanks for the clarifications provided by the authors.
> >
> > Overall I am satisfied with most of the responses.
> >
> > Regarding the experiments on adversarial attack, I still think that the drawn conclusions on the ability of the proposed method to achieve a higher success rate than SGD-based methods are questionable. I wonder if the adversarial attack is a good setting to demonstrate the practical effectiveness of the GH-methods. Indeed, the higher success rate is achieved at the cost of a higher L2 distortion. Hence comparing the succ. rates of the methods for different L2 error levels is non-informative. It will be more relevant to have a Pareto plot (success rate vs L2 error) issued from different $\lambda$ to effectively compare the methods.

---

> > > ### Author Response · Authors · 2022-08-09
> > > **Thank you for replying to our response**
> > >
> > > Thank you for the helpful suggestion. The Pareto plot of success rate v.s. L2 error will certainly provide useful information to compare the methods for different L2 error levels.
> > > We would like to add it to the final version.

---

### Official Review · Reviewer_WqLq · 2022-07-15

**Rating:** 6
**Confidence:** 2
**Soundness:** 3 good
**Presentation:** 3 good
**Contribution:** 3 good

**Summary:**

This paper studies the Gaussian homotopy method for finding stationary points for non-convex optimization problems. Novel deterministic and stochastic single loop Gaussian homotopy algorithms are proposed, where the decision variable $x$ and the smoothing parameter $t$ are simultaneously updated, in contrast to the double loop Gaussian homotopy algorithms in the existing literature. Also, the corresponding zeroth-order algorithms are developed based on zeroth-order estimators of gradient and Hessian. The convergence rates of the proposed algorithms are analyzed. Empirical evaluations are provided to corroborate the theory.

**Questions:**

1. For $\text{SLGH}_{\mathrm{d}}$, the update of $t_k$ depends on some small $\epsilon > 0$. What's the role of this $\epsilon$? What's the relationship between this $\epsilon$ and the one in the theorems as the error guarantee on the gradient norm?

2. Following the above question, it seems to me that the proof for Theorem 3.4 only works for $SLGH_r$.
This is because in line 83 in the appendix, the proof requires $\sum_{k=1}^T t_k \leq \sum_{k=1}^T t_1\gamma^{k-1}$, which is not necessarily true for $\text{SLGH}_{\mathrm{d}}$, as $t_k$ there is always at least $\epsilon$.

3. In the theorem statements, the $\hat x$ should be more specified. According to the proofs, the guarantee should be $\max_{k\in[T]} \|\nabla f(x_k)\|\leq \epsilon$.

4. From the intuition behind the Gaussian homotopy method, this kind of algorithm should perform better than the vanilla gradient descent because it is expected to find a better solution by smoothing the geometry. However, this is not reflected in the theory proved in this paper.

**Limitations:**

The authors have adequately addressed the limitations of their work.

**Strengths And Weaknesses:**

This paper is well-motivated and well-written, and overall the content is easy to follow. The idea to simultaneously update $(x,t)$ is intuitive and natural. The convergence analysis seems to be standard, and I don't see much novelty in this regard (maybe the authors can highlight if there is any special difficulty). The empirical study looks good to me.

---

> ### Author Response · Authors · 2022-08-02
> **Thank you for providing comments that will refine our paper. We address your detailed questions below.**
>
> Q1: For $\text{SLGH}_{\text{d}}$, the update of $t_k$ depends on some small $\epsilon>0$. What's the role of this  $\epsilon>0$? What's the relationship between this  $\epsilon>0$ and the one in the theorems as the error guarantee on the gradient norm?
>
> A1:
> - In the implementation of $\text{SLGH}_{\text{d}}$ without $\epsilon$, the value of $t$ can be nonpositive since it updates $t$ based on the derivative. We limit the range of values of $t$ using $\epsilon$ since the derivative of the GH function $F(x,t)$ cannot be defined at $t \leq 0$.
> - As Reviewer WqLq pointed out, both of the parameters used in the update rule of $t$ and that used in error guarantee in the theorems are denoted by $\epsilon$, but they are completely different. We fixed it in the revised version.
>
> Q2: Following the above question, it seems to me that the proof for Theorem 3.4 only works for $SLGH_r$ . This is because in line 83 in the appendix, the proof requires $\sum_{k=1}^T t_k \leq \sum_{k=1}^T t_1\gamma^{k-1}$, which is not necessarily true for $\text{SLGH}_{\text{d}}$, as $t_k$ there is always at least $\epsilon$.
>
> A2:
> - We needed to prove an additional inequality to complete the proof for $\text{SLGH}_{\text{d}}$. We added the proof in the revised version.
> - Let me briefly explain how we can modify the proof. From the update rule of $t_k$, we have
> $$
> \sum_{k=1}^T t_k \leq  \sum_{k=1}^T \max \{ t_1\gamma^{k-1} ,\epsilon \} \leq \sum_{k=1}^T t_1\gamma^{k-1} + \epsilon T
> $$
> Now, $\epsilon$ can take any positive value because we have introduced it only to guarantee the differentiability of the GH function $F(x,t)$. Thus, the right-hand side becomes $O(1)$ by taking $\epsilon$ sufficiently small.
>
> Q3: In the theorem statements, the $\hat{x}$ should be more specified. According to the proofs, the guarantee should be $\max_{k\in[T]} \| \nabla f(x_k) \| \leq \epsilon$.
>
> A3:
> - $\hat{x}$ is defined in the proof of each theorem since it is different by Theorems and the definition is somewhat complicated. Although we mentioned it in lines 199 and 251 in the main paper, it might still be difficult to find the corresponding definition sentences. Based on Reviewer WqLq’s opinion, we emphasized the definition sentences in the proofs in the revised version.
>
> Q4: From the intuition behind the Gaussian homotopy method, this kind of algorithm should perform better than the vanilla gradient descent because it is expected to find a better solution by smoothing the geometry. However, this is not reflected in the theory proved in this paper.
>
> A4:
> - As Reviewer WqLq pointed out, the theoretical guarantee is limited to the convergence in terms of the gradient norm. However, it is NP-hard to find a globally optimal solution in our problem setup [Pardalos et al., Quadratic programming with one negative eigenvalue is NP-hard, 1991]. Moreover, our analysis results are still important for the following reasons:
>     - Our method has the potential to outperform (ZO)GD-based algorithms since it may be able to find better solutions using smoothing.
>     - We can prove that the iteration complexity of our method equals corresponding (ZO)GD-based methods.
>     - The experimental results have demonstrated that our method can find better solutions than (ZO)GD-based algorithms in comparable computation time.

---

> > ### Comment · Reviewer_WqLq · 2022-08-06
> > **Thanks for the clarifications**
> >
> > Overall I'm satisfied with the authors' response.
> >
> > Regarding the $\hat x$ in the theorem statements, I think it's still meaningful to include its precise definition, especially as it is different for different algorithms. Otherwise the description like 'we can find $\hat x$' is a bit vague.

---

> > > ### Author Response · Authors · 2022-08-07
> > > **Thank you for replying to our response**
> > >
> > > We will add the definitions of $\hat{x}$ explained in the appendix in the main paper.

---

### Meta-Review · Area_Chair_Fnd4 · 2022-08-26

**Recommendation:** Accept
**Confidence:** Certain

**Metareview:**

All reviewers agree that it is a well-written paper.

**Award:**

No

---

### Decision · Program_Chairs · 2022-09-14

Accept